# Vangl2 acts at the interface between actin and N-cadherin to modulate mammalian neuronal outgrowth

Steve Dos-Santos Carvalho[1,2], Maite M Moreau[1,2], Yeri Esther Hien[1,2], Mikael Garcia[3,4], Nathalie Aubailly[1,2], Deborah J Henderson[5], Vincent Studer[3,4], Nathalie Sans[1,2], Olivier Thoumine[3,4†], Mireille Montcouquiol[1,2†*]

[1]INSERM, Neurocentre Magendie, U1215, Bordeaux, France; [2]Univ. Bordeaux, Neurocentre Magendie, U1215, Bordeaux, France; [3]CNRS, Interdisciplinary Institute for Neuroscience, UMR 5297, Bordeaux, France; [4]Univ. Bordeaux, Interdisciplinary Institute for Neuroscience, UMR 5297, Bordeaux, France; [5]Biosciences Institute, Newcastle University, Centre for Life, Newcastle upon Tyne, United Kingdom

*For correspondence:
mireille.montcouquiol@inserm.fr

†These authors contributed equally to this work

Competing interests: The authors declare that no competing interests exist.

**Abstract** Dynamic mechanical interactions between adhesion complexes and the cytoskeleton are essential for axon outgrowth and guidance. Whether planar cell polarity (PCP) proteins, which regulate cytoskeleton dynamics and appear necessary for some axon guidance, also mediate interactions with membrane adhesion is still unclear. Here we show that Vangl2 controls growth cone velocity by regulating the internal retrograde actin flow in an N-cadherin-dependent fashion. Single molecule tracking experiments show that the loss of *Vangl2* decreased fast-diffusing N-cadherin membrane molecules and increased confined N-cadherin trajectories. Using optically manipulated N-cadherin-coated microspheres, we correlated this behavior to a stronger mechanical coupling of N-cadherin with the actin cytoskeleton. Lastly, we show that the spatial distribution of Vangl2 within the growth cone is selectively affected by an N-cadherin-coated substrate. Altogether, our data show that Vangl2 acts as a negative regulator of axonal outgrowth by regulating the strength of the molecular clutch between N-cadherin and the actin cytoskeleton.

## Introduction

Core planar cell polarity (PCP) is a conserved signaling pathway known to regulate cytoskeleton dynamics in a large variety of cell types. PCP signaling has multiple roles in morphogenesis and patterning at both the single-cell and tissue levels, although the precise downstream effectors are still ill-defined (*Adler, 2012*; *Ezan and Montcouquiol, 2013*; *Goodrich, 2008*; *Montcouquiol and Kelley, 2019*; *Singh and Mlodzik, 2012*; *Tissir and Goffinet, 2013*; *Walck-Shannon and Hardin, 2014*; *Wallingford, 2012*). The original core PCP signaling cascade is composed of three transmembrane proteins called van gogh/strabismus (vang/stbm), flamingo/starry night (fmi/stan), frizzled (fz) and their mammalian orthologs Van gogh-like 1 and 2 (Vangl1/2); Cadherin/EGF/LAG seven-pass G-type receptor 1, 2, and 3 (Celsr1/2/3) and Frizzled 3 and 6 (Fzd3/6). Together with three other cytosolic proteins known as Dishevelled (Dsh/Dvl), Prickle and Diego (Dgo) as well as their mammalian orthologs Dishevelled 1, 2, and 3 (Dvl1/2/3); Prickle 1 and 2 (Pk1/2) and Diversin/Ankrd6, these proteins define a conserved PCP signaling cassette in epithelia from invertebrates to mammals (*Guo et al., 2004*; *Klein and Mlodzik, 2004*; *Singh and Mlodzik, 2012*). In mammals, including humans, mutations in these genes can lead to neural tube defects, which are sometimes associated with neonatal lethality (*Curtin et al., 2003*; *Montcouquiol et al., 2003*; *Torban et al., 2012*). Such severe defects have been reported in mice with single or double deletion of genes such as *Frizzled3/6* (*Wang et al., 2006*), *Vangl1* and/or *Vangl2* (*Song et al., 2010*), *Dvl1* and *Dvl2* (*Hamblet et al.,*

2002) and *Celsr1* (*Curtin et al., 2003*). Careful analysis of the brains of *Fzd3* and *Celsr3* single-mutant mice revealed strikingly similar defects in axonal tract formation, such as in the anterior commissure and the thalamocortical and corticothalamic tracts (*Tissir et al., 2005*; *Tissir and Goffinet, 2013*; *Wang et al., 2006*; *Zhou et al., 2008*). Both genes are also required for the guidance of monoaminergic axons along the anterior-posterior axis and in the anterior turning of commissural axons in the spinal cord (*Fenstermaker et al., 2010*; *Lyuksyutova et al., 2003*). Overall, these studies suggested that *Celsr3* and *Fzd3* deletion disrupts the ability of the growth cones to respond to guidance cues but not outgrowth (*Chai et al., 2014*; *Hua et al., 2014*; *Song et al., 2010*). This axonal guidance function of Celsr3 and Fzd3 was extended to Vangl2 when some studies also reported apparent axonal guidance deficits in a mouse model carrying a spontaneous missense mutation for *Vangl2* (called Loop-tail or Vangl2^Lp) (*Fenstermaker et al., 2010*; *Onishi et al., 2013*; *Shafer et al., 2011*). However, more recent papers using a conditional mutant for both the *Vangl1* and *Vangl2* genes failed to report such deficits, suggesting that Vangl2 might not share the same global molecular properties as Celsr3 and Fzd3 in young neurons of the developing brain (*Qu et al., 2014*).

Core PCP proteins are critical during the intense developmental periods of tissues' growth and remodelling as illustrated during posterior body formation in zebrafish (*Harrington et al., 2007*) or germband extension in *Drosophila* (*Zallen, 2007*). In both cases, there is a strong interplay between core PCP proteins, the dynamics of the cytoskeleton and the spatial regulation of adhesion molecules (*Ciruna et al., 2006*; *Classen et al., 2005*; *Dohn et al., 2013*; *Heller and Fuchs, 2015*; *Jessen and Jessen, 2017*; *Warrington et al., 2013*; *Yin et al., 2008*). It was in fact suggested that PCP signaling controlled the assembly/disassembly of adherens junctions during Drosophila wing epithelium growth (*Classen et al., 2005*). More recently it was demonstrated that deletion of core PCP genes increases both the amount and the stable fraction of E-cadherin, notably through a decrease in the efficiency of E-cadherin recycling (*Warrington et al., 2013*). This is believed to occur through the formation of puncta and signalosome-like structures of core PCP proteins at epithelial cell junctions (*Strutt et al., 2011*; *Strutt et al., 2016*; *Warrington et al., 2013*; *Warrington et al., 2017*). In the developing brain, the dynamic assembly/disassembly of adhesion complexes is also necessary for neuronal outgrowth. In essence, the entire adhesion apparatus is recycled/reused to fit the moment-by-moment demands as well as to adapt to the environment encountered during movement (*Vitriol and Zheng, 2012*). Adhesion complexes at the growth cone membrane can serve as a link between the substrate and the internal retrograde F-actin flow, leading to neuronal outgrowth via a mechanism known as the 'molecular clutch' which postulates that increased adhesion attenuates retrograde actin flow, resulting in net protrusion and cell motility (*Lin and Forscher, 1995*; *Mitchison and Kirschner, 1988*; *Suter and Forscher, 1998*).

Here, we postulated that Vangl2 participates in neuronal outgrowth in the developing brain by affecting adhesion molecules and cytoskeleton dynamics within the growth cone. Using a combination of biophysical methods and live imaging approaches, we show that Vangl2 regulates the dynamics of N-cadherin at the plasma membrane, thereby negatively controlling the engagement of N-cadherin adhesions with the retrograde actin flow and, as a result, slowing down neuronal outgrowth. Overall, our study highlights a novel molecular function of Vangl2 in early hippocampal development and defines Vangl2 as a neuronal molecular brake in young neurons.

## Results

### Early *Vangl2* deletion leads to partial corpus callosum and hippocampal commissure agenesis

To evaluate the impact of Vangl2 deletion in the forebrain, we generated Emx1-Cre*Vangl2* cKO conditional mutants, efficiently deleting *Vangl2* in the early dorsal telencephalon (V*angl2* cKO mice). As expected, Vangl2 expression did not vary in the cerebellum or spinal cord, where Emx1 is not expressed, but was significantly decreased in the telencephalon as early as E13.5, including the hippocampus and cortex (*Figure 1—figure supplement 1A,B*). In these early stages, we did not observe a significant disruption of the internal capsule fibers (*Figure 1—figure supplement 1C*). Analysis of brain sections from *Vangl2* cKO mice confirmed the agenesis of both the corpus callosum (cc, magenta outline) and the hippocampal commissure (hc, cyan outline) in the dorso-caudal region of the forebrain, while other major commissures appeared intact, as previously reported (*Qu et al.,*

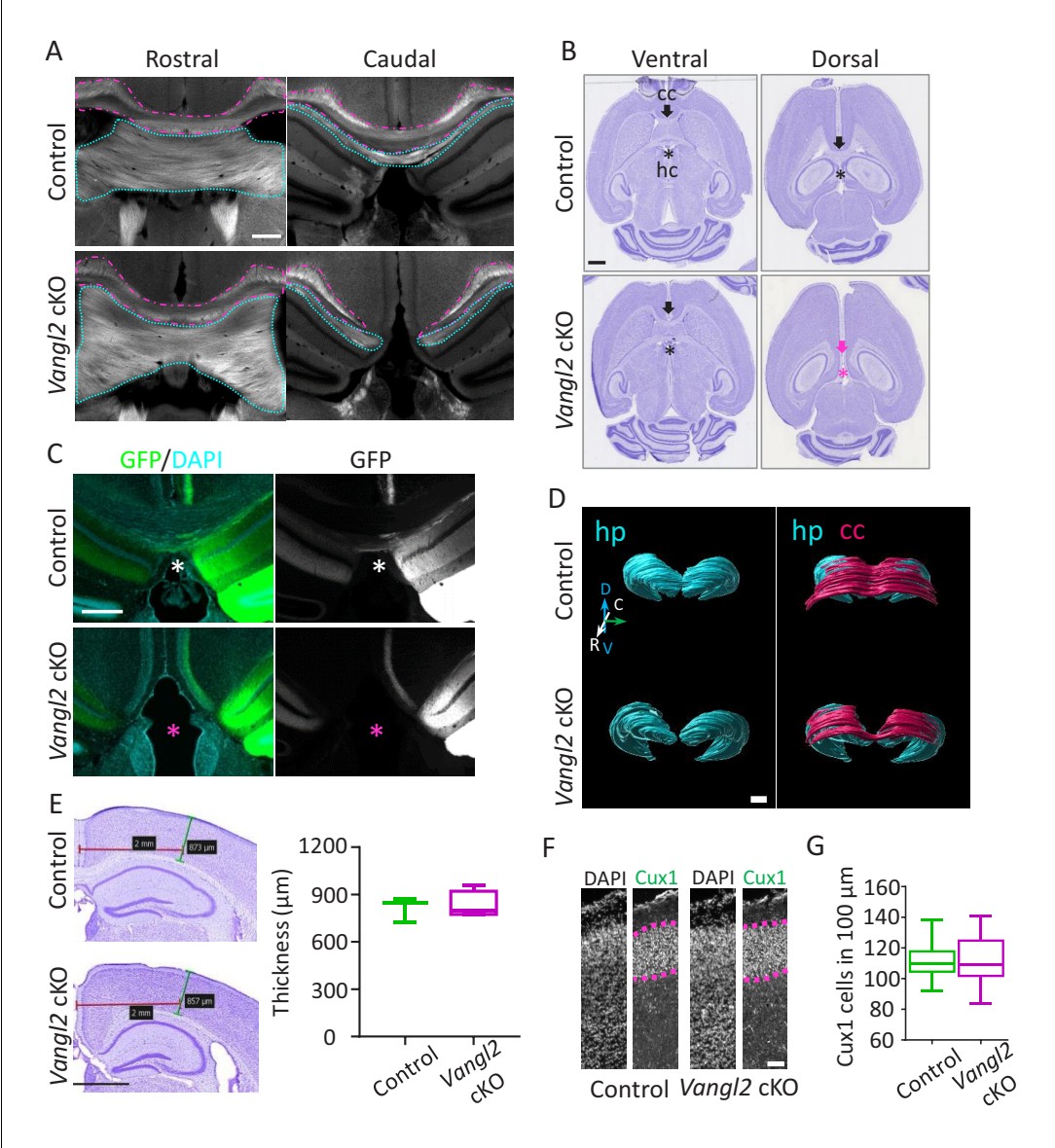

**Figure 1.** Vangl2 affects commissural tracts. (**A**) Immunostaining with FluoroMyelin of coronal sections from P21 control (upper panel) and *Vangl2* cKO (lower panel) brains in a rostral (left) and more caudal (right) location. Note the absence of the corpus callosum (magenta outline) and hippocampal commissure (cyan outline) caudally in the *Vangl2* cKO. Scale bar: 100 µm; n = 3 independent experiments. (**B**) Cresyl Violet staining of horizontal sections from P31 control (upper panel) and *Vangl2* cKO brains (lower panel). Note that the corpus callosum (cc, magenta arrow) and the hippocampal commissure (hc, magenta asterisk) are absent dorsally in *Vangl2* cKO brains.Scale bar: 1 mm. (**C**) Coronal sections from 22-week-old control (upper panel) and *Vangl2* cKO (lower panel) brains, injected with an AAV-GFP virus and labeled with DAPI (cyan). Note that hippocampal fibers (in green) cross the cerebral midline in control mice but not in the mice lacking Vangl2. Scale bar: 50 µm. (**D**) 3D rendering of the hippocampus and the cc covering the hippocampus area. The reconstruction shows the absence of the splenium corpus callosum area in the *Vangl2* cKO mouse (white arrow). Abbreviations: hippocampus (hp), corpus callosum (cc), dorso/ventral (D/V), rostrao/caudal (R/C). Scale bars: 800 µm. (**E**) Cresyl violet staining of 10 week-old control and *Vangl2* cKO brains and quantification of cortical thickness of the somato-sensory area. Scale bar: 1 mm; n = 3–5 mice. Data are presented as box-and-whisker plots (min/max) based on three independent experiments. (**F**) Cux1/DAPI staining of P0 cortices from control and *Vangl2* cKO mice. The Cux1 layer is depicted with a dotted magenta line. Scale bar: 100 µm. (**G**) Quantification analysis show no difference in the number of Cux1 positive cells between P0 control and *Vangl2* cKO mice; n = 3 independent experiments. Data are presented as box-and-whisker plots (min/max) based on three independent experiments.

The online version of this article includes the following figure supplement(s) for figure 1:

**Figure supplement 1.** Vangl2 affects commissural tracts.

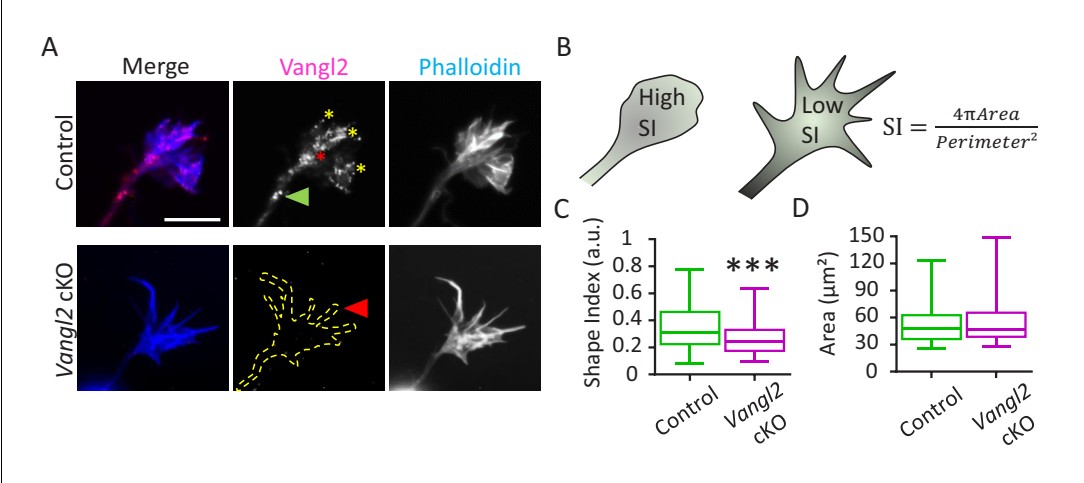

**Figure 2.** Vangl2 affects growth cone morphology. (A) Cultured hippocampal neurons from control and *Vangl2* cKO mice at DIV2 on laminin substrate, labeled with an anti-Vangl2 antibody (magenta) and phalloidin (blue). The dotted lines indicate the shape outline. Scale bar: 5 μm. (B) The Shape Index (S.I.) was calculated by dividing the growth cone's area by its perimeter (see Materials and methods). Growth cones that are more lamellipodia-like tend to present S.I. values larger than growth cones that have more filopodial shape. (C) Shape index (S.I.) analysis of control and *Vangl2* cKO growth cones from neurons plated on laminin substrate. The S.I. is defined as $4\pi Area/Perimeter^2$; n = 54–60 neurons. (D) Average area of growth cones from control and *Vangl2* cKO neurons. Data are presented as box-and-whisker plots (min/max) based on 3 (C, D) independent experiments; ***p<0.001 by the Mann-Whitney test (C).

*2014*) (*Figure 1A,B* and *Figure 1—figure supplement 1D,E*). To further validate the hc deficit, we injected AAV9 viruses coding for GFP in the CA3 region of the hippocampus and processed the brain 4 weeks later. The results showed that in control mice, commissural colossal and hippocampal fibers cross the cerebral midline, while in *Vangl2* cKO mice they did not (*Figure 1C*). Three-D rendering representation of the hippocampus and cc created from coronal stack images stained with cresyl violet demonstrated the matching deficit of cc crossing above the hippocampus in *Vangl2* cKO mice (*Figure 1D*). The observed deficit in cc was not due to a decrease in upper-layer neurons, as the thickness of the somato-sensory region of the cortex, or the number of Cux1-positive neurons and the thickness of the layer they define, were unchanged between controls and *Vangl2* cKO (*Figure 1E–G*). Alltogether our data suggest that an early deletion of *Vangl2* in the telencephalon leads to cc and hc commissural deficits restricted to the dorso-caudal region. We next sought to evaluate if such a phenotype could be the result of abnormal axonal outgrowth in the absence of Vangl2.

### *Vangl2* deletion increases neuronal outgrowth on N-cadherin-coated substrates

Immunofluorescence labeling of cultured hippocampal neurons showed the presence of Vangl2 (magenta) in neuronal growth cones at 2 days in vitro (DIV2) in relation to F-actin distribution (phalloidin staining, blue) (*Figure 2A*). Vangl2 protein appeared in the form of puncta, which were located along filopodia (yellow asterisks), close to the center of the growth cones (red asterisk), and along the growing neurite shafts (green arrowhead), but also exhibited some degree of diffuse staining. Similar punctate labeling was reported for core PCP proteins during highly dynamic processes, such as vertebrate gastrulation, where transient asymmetric localization of polarity proteins was observed (*Ciruna et al., 2006*; *Yin et al., 2008*). In *Vangl2* cKO neurons, most of the staining disappeared (*Figure 2A*), cross-validating the antibody specificity and the efficiency of *Vangl2* deletion. We noticed that the absence of Vangl2 affected growth cone morphology (*Figure 2A–C*). To quantify this effect, we defined a normalized shape index (S.I.), where values close to one represent circular objects and values close to zero represent elongated ones (*Figure 2B*). S.I. values were significantly smaller in the growth cones of *Vangl2* cKO mice than in those of controls, confirming that the former group exhibits more fusiform-shaped growth cones than the latter group. The

average area of the growth cones, however, was unchanged between controls and *Vangl2* cKOs (*Figure 2D*).

Since growth cone morphology strongly relies on both substrate adhesion and cytoskeleton dynamics, we next examined the impact of Vangl2 on these two parameters. We first evaluated whether Vangl2 affected the net motility of growth cones, depending on substrate coating. To this end, we used time-lapse differential interference contrast (DIC) microscopy to track the movement of growth cones from hippocampal neurons of control and *Vangl2* cKO mice cultured for 2 DIV on glass substrates coated with N-cadherin-Fc (Ncad-Fc). On average, within a 45 min observation period, growth cones from *Vangl2* cKO mice plated on Ncad-Fc covered a longer distance (*Figure 3A*, dotted pink line, see also *Videos 1* and *2*) and migrated faster than controls (+73%, *Figure 3B*). The growth cones of *Vangl2* cKO neurons paused less in total (*Figure 3C*) and had a shorter average time per pause than controls (*Figure 3D*), but the velocity between pauses was also higher than that of control growth cones (*Figure 3E*). Pauses in displacement over time were defined as no movement of the centroid of the growth cone for 1 min (*Figure 3F*).

To examine the influence of the substrate coating on these processes, we repeated these experiments with neurons plated on laminin, a neuronal growth-promoting substrate, or poly-L-lysine (PLL). We found no significant difference in growth cone movement between *Vangl2* cKO and control neurons on laminin or PLL (*Figure 3G,J,K*), and no difference in the total time spent pausing was observed on laminin-coated substrates (*Figure 3H*). Altogether, these results show that *Vangl2* deletion causes a selective enhancement of growth cone velocity which is dependent on N-cadherin adhesion. To evaluate whether Vangl2 is implicated in the neuronal response to growth and/or guidance factors, we cultured control and *Vangl2* cKO neurons in the presence of Netrin-1 or Wnt5 at 1 DIV on a laminin-coated substrate. After 24 hr, we evaluated axonal length by immunolabeling Tau, a microtubule-associated protein expressed only in axons. The addition of either Netrin-1 or Wnt5 led to a significant increase in axonal length in both control and *Vangl2* cKO neurons, and this increase was comparable between genotypes (*Figure 3I*). This demonstrates that neuronal outgrowth in response to guidance cues is not affected by the absence of Vangl2.

## *Vangl2* deletion decreases retrograde actin flow in growth cones

Given that the dynamic mechanical coupling between N-cadherin adhesion and retrograde actin flow regulates growth cone migration in hippocampal neurons (*Bard et al., 2008*; *Garcia et al., 2015*), we hypothesized that *Vangl2* deletion could affect these processes. We first measured the actin flow rate in the growth cones of *Vangl2* cKO or control neurons by detecting individual actin-mEos2 molecules using single particle tracking combined with Photoactivated Localization Microscopy (sptPALM) under total internal reflection fluorescence (TIRF) illumination to focus on molecules in close proximity to the substrate (*Figure 4A*, *Videos 3* and *4*). At a frame rate of 2 Hz, we were able to isolate slowly moving actin-mEos2 molecules incorporated in the filamentous actin network while excluding fast-diffusing actin monomers. The reconstructed trajectories lasted a few seconds (median = 4.5 s). The shape of the mean squared displacement (MSD) of the trajectories over time (t), as fitted by the function $4Dt^{\alpha}$, allowed a classification of the trajectories (*Figure 4—figure supplement 1A*), that is high $\alpha$ values correspond to directed motion, intermediate $\alpha$ values to Brownian motion, and low $\alpha$ values to confined motion (*Garcia et al., 2015*) (*Figure 4—figure supplement 1B,C,D*). We first validated this analysis for growth cones from mice hippocampal neurons plated on PLL substrates, which are well spread and exhibit strong and homogeneous actin flow (*Figure 4—figure supplement 1E,F*, *Video 5*). The calculated actin flow rates were comparable to the ones obtained by kymograph analysis (*Figure 4—figure supplement 1I,J*). However, when neurons were plated on growth-promoting substrates such as Ncad-Fc, they adopted more fusiform morphologies associated with a reduction of their projected surface area, and migrated faster compared to growth cones plated on PLL (*Figure 4—figure supplement 1G,H*) (*Garcia et al., 2015*). In that situation, the actin flow is more erratic and of lower amplitude (*Figure 4*).

On Ncad-Fc substrates, the variety of actin-mEos2 dynamics was highlighted by plotting the distribution of $\alpha$ coefficients obtained for control and *Vangl2* cKO neurons (*Figure 4B,C*, compare also with PLL substrate in *Figure 4—figure supplement 2K*). In the absence of Vangl2, neurons presented a shift from more directed actin trajectories (higher $\alpha$ values) towards less diffuse/more confined trajectories (lower $\alpha$ values) when compared with controls (*Figure 4D*). Measurements restricted to directed trajectories (*i.e.* $\alpha \geq 1.5$) showed that the velocity of actin-mEos2 molecules

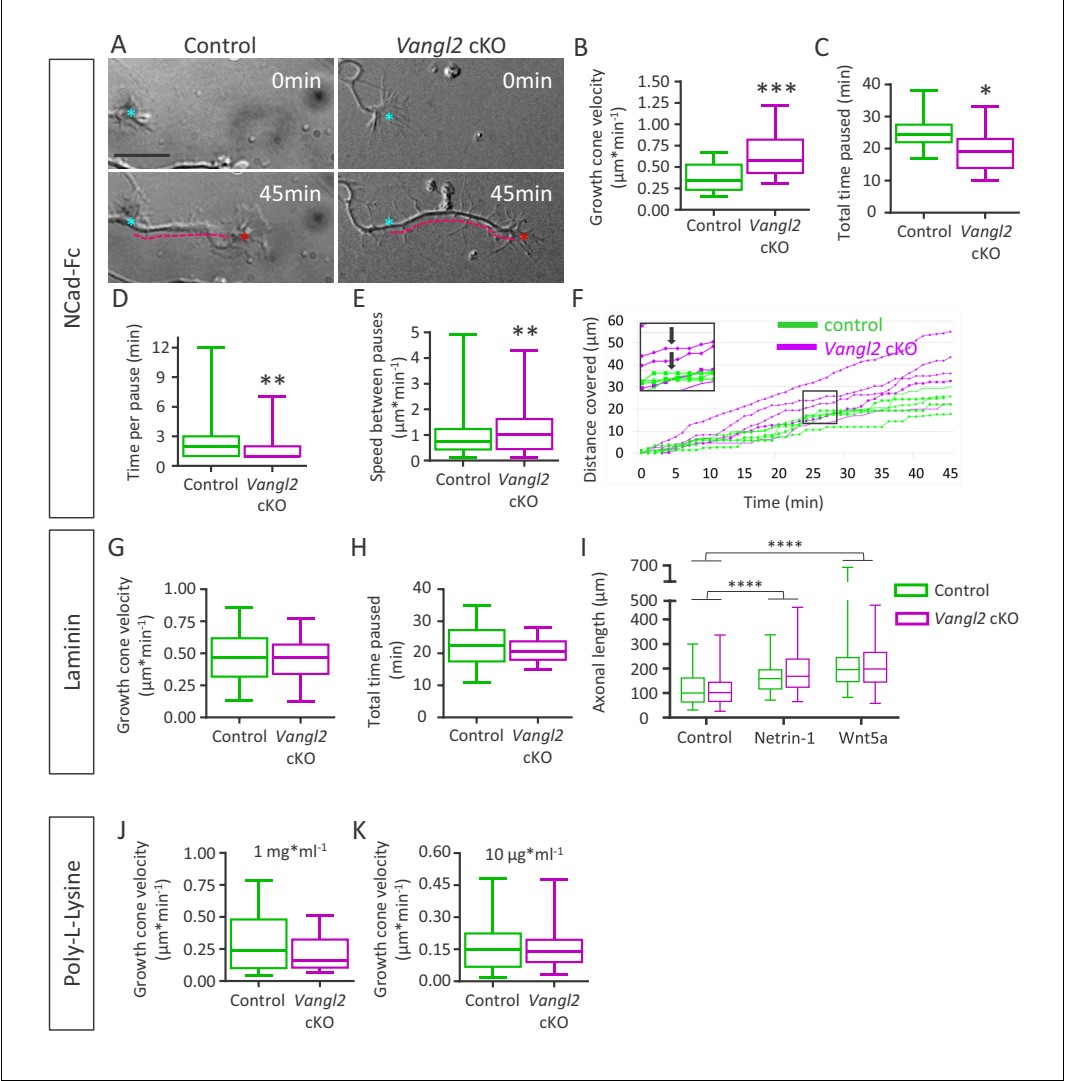

**Figure 3.** Deletion of *Vangl2* increases growth cone velocity on Ncad-Fc substrate. (A) Representative images of neurite extension from t = 0 min (cyan asterisk) to t = 45 min (magenta dotted line) at DIV2 in *Vangl2* cKO and control neurons cultured on Ncad-Fc-coated substrates. Scale bar, 10 µm. (B) Quantification of the average growth cone velocity in control and *Vangl2* cKO neurons; n = 24–28 neurons. (C) Quantification of total time spent in pauses during a 45 min period for *Vangl2* cKO and control growth cones. (D) Quantification of the time per pause in *Vangl2* cKO neurons and controls. (E) Quantification of growth cone speed (excluding pauses) in *Vangl2* cKO and control growth cones. For C, D, E: n = 10–11 neurons. (F) Graph of the cumulative distance covered by four representative individual *Vangl2* cKO (purple) and control (green) growth cones over time. A pause was defined as no movement of the centroid of a growth cone for one minute or more (black arrows). (G, H) Average growth cone velocity and total pause duration in control and *Vangl2* cKO neurons on laminin substrates. n = 32–40 neurons (for total time paused, n = 16–20 neurons). (I) Bar graph showing the axonal lengths of both control (n = 94) and *Vangl2* cKO (n = 84) neurons in the control condition or treated with Netrin-1 or Wnt5a. (J, K) Average growth cone velocity and total pause duration in control and *Vangl2* cKO neurons on PLL substrates. PLL at 1 mg\*ml$^{-1}$n = 21–23 neurons; PLL at 10 mg\*ml$^{-1}$: n = 25–27 neurons. Data are presented as box-and-whisker plots (min/max) based on 3 (B–E, I) or 2 (G, H, J, K) independent experiments; \*p<0.05, \*\*p<0.01 or \*\*\*p<0.001, by Student's t-test (B, C, G, H, J), the Mann-Whitney test (D, E, K), or 2-way ANOVA followed by Tukey's multiple-comparisons test (I).

within actin filaments was, on average, 23% slower in the growth cones of *Vangl2* cKO neurons than in control growth cones (*Figure 4E*), suggesting that in the absence of Vangl2, the speed of retrograde actin flow in growth cones slowed down. These results are consistent with a mechanism known as the 'molecular clutch', which postulates that increased adhesion attenuates retrograde actin flow, resulting in net protrusion and cell motility (*Lin and Forscher, 1995*; *Mitchison and Kirschner, 1988*; Suter and Forscher, 1995). This hypothesis was supported by the absence of a global deficit in actin polymerization levels, as assessed by a similar ratio of F- to G-actin (F/G) in neurons with and

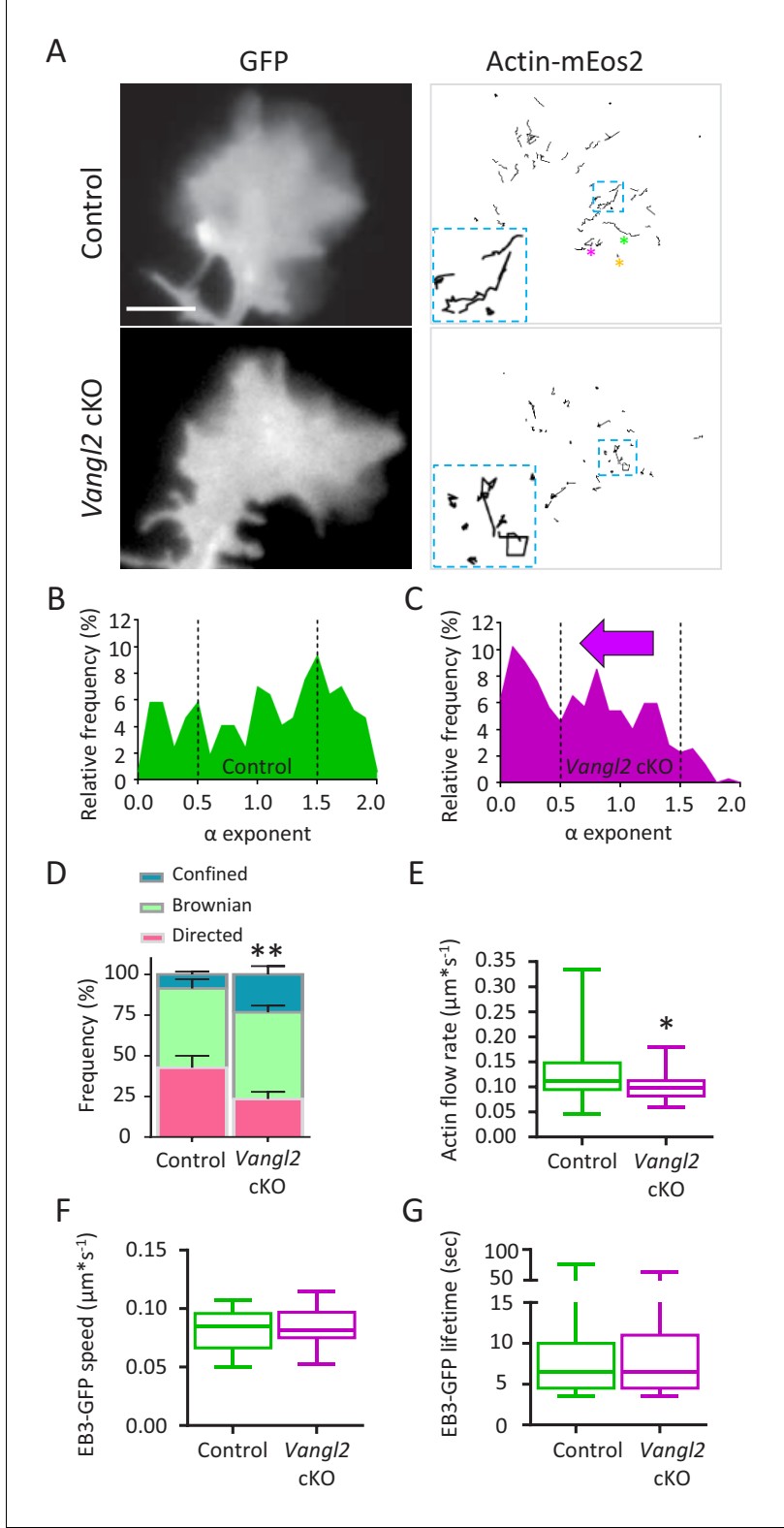

**Figure 4.** Deletion of Vangl2 decreases actin treadmilling. (**A**) Representative images of growth cones from control and Vangl2 cKO neurons expressing GFP and actin-mEos2. Images show the growth cone filled with GFP (left) and the trajectories of single actin-mEos2 molecules (in black) recorded over a 3 min period at 2 Hz (right). Insets show higher-magnification examples of the variability in individual trajectories. Scale bar, 5 µm. (**B, C**) Distribution of actin-mEos2 molecules α values in control and Vangl2 cKO neurons. n = 184–440 trajectories, 6–7 neurons. (**D**)

*Figure 4 continued on next page*

*Figure 4 continued*

Frequency distribution of directed, Brownian and confined trajectories of actin-mEos2 molecules in control and Vangl2 cKO neurons plated on Ncad-Fc substrates. (E) Speed of retrograde actin flow as extracted from the directed trajectories in controls and Vangl2 cKO neurons. n = 73–79 trajectories, 6–7 neurons. n = 8 sister cultures (DIV3) from eight different mice. (F, G) Quantification of the speed and lifetime of EB3-GFP particles in control and Vangl2 cKO neurons. n = 18–19 neurons. Data are presented as box-and-whisker plots (min/max) based on three independent experiments; *p<0.05, **p<0.01 by the Mann-Whitney test (D, E).

The online version of this article includes the following figure supplement(s) for figure 4:

**Figure supplement 1.** SptPALM is an accurate method for the analysis of F-actin flow in growth cones.

**Figure supplement 2.** Theoretical prediction of the mechanical coupling between the actin flow and transmembrane adhesion.

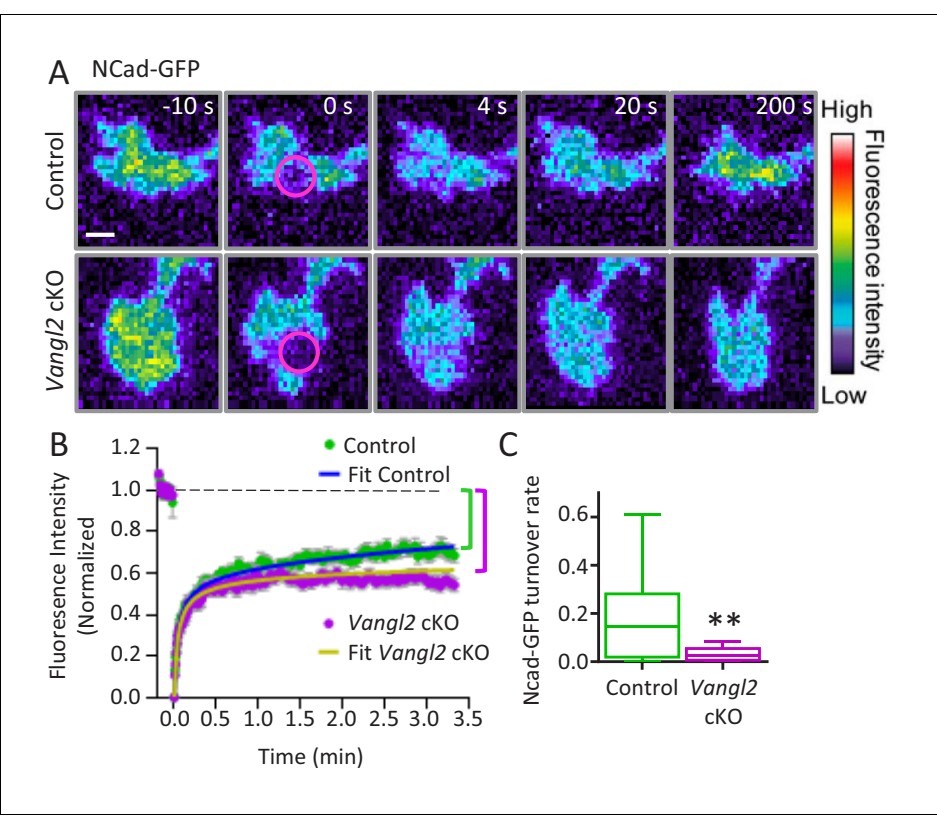

**Figure 5.** Vangl2 controls Ncad-GFP turnover in growth cones. (A) Representative images of the recovery of Ncad-GFP after photobleaching, recorded by TIRF illumination in growth cones from controls (upper panels) and Vangl2 cKO (lower panels) neurons plated on Ncad-Fc substrates. Growth cones were photobleached in the selected area (circle, time 0 s) and the fluorescence recovery was recorded for the following 200 s (color coding). Scale bar, 5 μm. (B) Mean FRAP curves of the Ncad-GFP signal over time, containing individual data points ± SEM for each experiment. In the absence of Vangl2, the initial recovery (20 s) is similar to that of the control, but the long-term recovery is significantly lower. Solid lines represent a fitted diffusion-reaction model. (C) Quantification of the turnover rate of Ncad-GFP molecules involved in homophilic bonds at the membrane shows that protein turnover is significantly reduced (83%) in Vangl2 cKO growth cones compared to controls. n = 13–14 neurons, three experiments. Data are presented as box-and-whisker plots (min/max) based on three independent experiments; **p<0.01 by Student's t-test (C).

The online version of this article includes the following figure supplement(s) for figure 5:

**Figure supplement 1.** *Vangl2* deletion does not affect N-cadherin expression in the hippocampus, nor the proportion of freely diffusing or confined N-cadherin at the membrane.

without Vangl2 (*Figure 4—figure supplement 2L*). Microtubule dynamics, which is also important for neuronal outgrowth and guidance, was unaffected in the growth cones of *Vangl2* cKO neurons, as assessed by tracking the movement of EB3-GFP comets (*Figure 4F,G*, *Videos 6* and *7*), suggesting that Vangl2 primarily affects F-actin dynamics.

To better interpret the sptPALM data, we ran computer simulations of actin dynamics in virtual growth cones (*Figure 4—figure supplement 2A–F*), taking into account the peripheral incorporation of diffusing actin monomers into a filamentous network that slowly moves rearward and can transiently couple to transmembrane adhesions at the substrate level. Given realistic parameters, the simulations closely mimicked the sptPALM data (*Figure 4—figure supplement 2C*) and predicted that the fraction of directed trajectories is affected (i) by the coupling strength between the actin flow and substrate adhesions (*Figure 4—figure supplement 2D,E*), but also (ii) by the actin velocity itself (*Figure 4—figure supplement 2F*). The observed shift towards smaller α values in *Vangl2* cKO neurons (*Figure 4C,D*) suggests that the absence of this PCP protein leads to an increase in the coupling strength between the actin flow and adhesion molecules, thereby reducing the actin flow rate and increasing the migration speed.

## Vangl2 controls N-cadherin dynamics at the membrane

Since we observed an increase in axonal outgrowth caused by *Vangl2* deletion only on Ncad-Fc substrates, we further investigated the relationship between Vangl2 and N-cadherin. We did not find differences in the levels of N-cadherin or β-catenin expression in hippocampi from controls and *Vangl2* cKO mice, suggesting that *Vangl2* deletion does not affect the global expression of the N-cadherin/catenin complex (*Figure 5—figure supplement 1*). To examine whether Vangl2 regulates the levels of N-cadherin at the growth cone membrane, we carried out FRAP experiments on N-cadherin-GFP (Ncad-GFP)-expressing neurons plated on Ncad-Fc coated glass under TIRF illumination. The recovery curve of Ncad-GFP showed a typical biphasic profile (*Figure 5A,B*). In both control and *Vangl2* cKO neurons, approximately 50% of fluorescence was recovered in a rapid phase (~20 s) that represents the replenishment of freely diffusing Ncad-GFP molecules at the plasma membrane (*Figure 5B*). At the end of the second phase, which had much slower dynamics (~200 s), control neurons recovered up to 68% of their fluorescence, while *Vangl2* cKO growth cones recovered only to 54%. Therefore, in the absence of Vangl2, there was an increased fraction of slow Ncad-GFP molecules at the membrane (+14%) or an extended turnover time of Ncad-GFP molecules engaged in homophilic bonds (*Bard et al., 2008*; *Garcia et al., 2015*). The FRAP data were fitted with a diffusion/reaction model, from which we extracted an 83% decrease in the long-term turnover rate of N-cadherin adhesions in the absence of Vangl2, supporting our hypothesis (*Figure 5C*). These results suggest that, in the absence of Vangl2, a significant fraction of N-cadherin with low mobility or stable anchoring is available for engagement in the clutch mechanism.

## *Vangl2* deletion decreases N-cadherin diffusion

To better understand the heterogeneous behavior of N-cadherin at the membrane, we tracked individual N-cadherin-mEos2 (Ncad-mEos2) molecules using sptPALM coupled to TIRF at a high frame rate (20 Hz) in the periphery of the growth cone, where the actin treadmill occurs. Tracking individual molecules allowed us to generate maps showing their trajectories (*Figure 6A*, *Videos 8* and *9*). The MSDs calculated from reconstructed trajectories were fitted by the function 4Dt to yield the diffusion coefficient D (in $\mu m^2 * s^{-1}$). Molecules with values below D = 0.015 $\mu m^2 * s^{-1}$ (log(D)=−1.8, corresponding to the localization accuracy of our system for the detection of N-cad-mEos2) were considered as confined (gray area), while those with values above this threshold were considered as mobile (*Figure 6B,C*). The median diffusion coefficient of Ncad-mEos2 molecules was not different in growth cones from *Vangl2* cKO origin compared to controls (p=0.08) (*Figure 6—figure supplement 1A*). However, a comparison of the distribution of diffusion coefficients between genotypes showed a significant decrease in the proportion of fast-diffusing N-cadherin molecules in the absence of Vangl2, concomitant with a decrease in the fraction of confined molecules (*Figure 6B,C*). This population of slowly moving Ncad-mEos2 probably represents molecules engaged in durable interactions, which are likely to correspond to the larger fraction of confined molecules that was observed in FRAP experiments. A plot of the MSD of Ncad-mEos2 molecules over time confirmed a global

reduction in N-cadherin diffusion and covered area in the absence of Vangl2, with a sharp decrease after ~200 ms (**Figure 6D**).

## *Vangl2* deletion affects N-cadherin confinement domains

To better quantify the effect of *Vangl2* deletion on the proportion of Ncad-mEos2 molecules with slow or confined trajectories, we used image analysis to reconstruct super-resolution images of Ncad-mEos2 molecules in growth cones and evaluated the formation and number of 'nanometric' confinement domains (referred to as 'domains' in the rest of the manuscript) formed by slowly moving molecules (**Figure 6A**, right panels). The average size of those domains did not change in the absence of Vangl2 (**Figure 6—figure supplement 1C**), but its distribution showed a large peak between 80 to 120 nm and above 220 nm in the absence of Vangl2 (**Figure 6E**). Thus, in the absence of Vangl2, the diffusion of Ncad-mEos2 molecules is reduced, and the domain size of the slow/confined trajectories decreases. The strongest difference observed was a significant increase (+100%) in the density of domains per unit area (domains*$\mu m^{-2}$) in *Vangl2* cKO neurons (**Figure 6F**), confirmed by the frequency distribution of nanodomain density (**Figure 6G**). Altogether, our data suggest that in the absence of Vangl2, N-cadherin turnover and the number of fast-diffusing N-cadherin molecules at the membrane decrease, while the number of confined molecules increases. These results are consistent with the formation of *trans*-dimers of N-cadherin and/or an increased local engagement of N-cadherin with F-actin.

## N-cadherin substrate affects Vangl2 distribution

PCP is a process known to act both cell-autonomously and non-cell-autonomously (**Adler, 2012**; **Montcouquiol and Kelley, 2019**). Since Vangl2 affects N-cadherin behavior in neurons, we evaluated whether the substrate could reciprocally affect Vangl2 mobility at the membrane. For this purpose, we monitored mEos2-Vangl2 molecules on Ncad-Fc or laminin substrates using sptPALM coupled to TIRF, as before (**Figure 7A,B**). In growth cones, mEos2-Vangl2 molecules displayed a variety of diffusion phenotypes, with a large proportion of fast-moving molecules (**Figure 7C**). Molecules with values below D = 0.025 $\mu m^2$*$s^{-1}$ (log(D)=−1.6, corresponding to the localization accuracy of our system for the detection of mEos2- Vangl2), were considered as confined (gray area), while those with values above this threshold were considered as mobile (**Figure 7C**). We observed no significant difference in the global median diffusion coefficient of Vangl2 molecules whether the neurons were grown on Ncad-Fc- or laminin-coated substrates (**Figure 7D**). When we sorted the trajectories to compare the diffusion coefficients between the periphery and the center of the growth cones, we observed a small decrease in Vangl2 diffusion at the periphery on both substrates (**Figure 7E,G**). Plots of the MSDs of mEos2-Vangl2 molecules over time show a clear reduction in the area covered by Vangl2 molecules at the periphery on both substrates, with a more pronounced difference on Ncad-Fc than on laminin (**Figure 7F,H**). If the average size of the mEos2-Vangl2 domains (nanometric confinement domains) was not different between the two types of substrates (**Figure 7I**), the domain size distribution revealed a slight shift towards a larger domain size on Ncad-Fc substrates compared with laminin (**Figure 7J**). The density of Vangl2 domains was higher on Ncad-Fc substrate than on laminin (**Figure 7K,L**), a difference that can be explained by the larger surface area of the growth cones grown on laminin-coated substrates (**Figure 7A,B**). Despite this difference in size, Vangl2 domains appeared to be more numerous at the periphery of growth cones cultured on Ncad-Fc substrates than at the periphery of those cultured on laminin (**Figure 7A**, right panel). When we calculated the ratio of mEos2-Vangl2 domains at the periphery vs the entire growth cone, we observed a significant increase (+41%) in this ratio on Ncad-Fc substrates compared to laminin substrates (**Figure 7M**). These data suggest that N-cadherin interacting in trans can affect the distribution of Vangl2 molecules.

## *Vangl2* deletion increases the mechanical coupling between N-cadherin adhesion and the actin flow

To characterize the physical coupling between N-cadherin adhesions and the underlying actin flow, we used Ncad-Fc-coated microspheres manipulated with optical tweezers. Microspheres were placed on the dorsal surface of growth cones, and the optical trap was continuously applied for 2 min (**Figure 8A**, **Videos 10** and **11**). In this paradigm, traction forces exerted by the retrograde

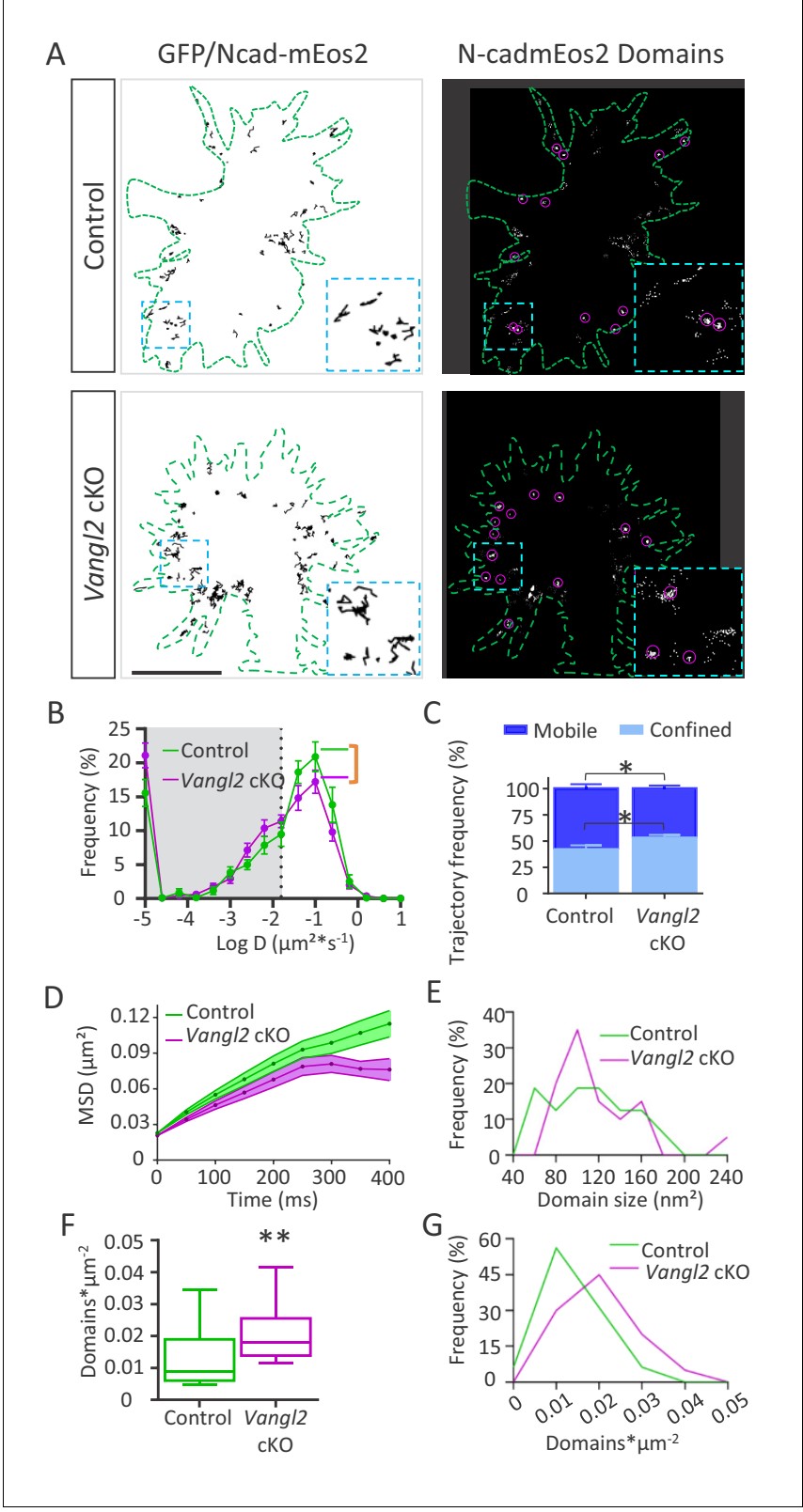

**Figure 6.** Ncad-mEos2 diffusion and domain density are altered in the absence of Vangl2. (**A**) Upper panel: on the left, representative image of single Ncad-mEos2 molecules trajectories (in black) detected over a 100 s period, superimposed on the GFP-based outline of the growth cone (green); on the right, corresponding super-resolution projections of all the single N-cadherin-mEos2 molecules detected. Lower panel: Similar illustrations for *Vangl2*

*Figure 6 continued*

cKO neurons expressing GFP and Ncad-mEos2. Scale bar, 2.5 µm. (B) Frequency distribution of diffusion coefficients on a logarithmic scale for control (green) and *Vangl2* cKO (magenta) neurons. n = 921–1389 trajectories, 11–19 neurons. (C) Quantification of confined and mobile Ncad-mEos2 particles in control and *Vangl2* cKO neurons. (D) Plot of the MSD at the periphery of growth cones as a function of time. (E) The frequency distribution of domain size at the growth cone periphery shows no significant differences between control and *Vangl2* cKO neurons. (F) Density of Ncad-mEos2 domains at the growth cone periphery in control and *Vangl2* cKO neurons. n = 16–20 neurons. (G) Frequency distribution of domain density at the growth cone periphery. Data are presented as box-and-whisker plots (min/max) from six different mice per genotype and three experiments. *p<0.05 and **p<0.01 by Student's t-test (C) or the Mann-Whitney test (F).

The online version of this article includes the following figure supplement(s) for figure 6:

**Figure supplement 1.** Frequency and cumulative distribution of Ncad-mEos2 molecules in growth cones.

F-actin flow are transmitted to the microsphere by endogenous transmembrane N-cadherin receptors, resulting in retrograde bead motion at the growth cone surface (*Bard et al., 2008*). After proper calibration, we estimated a minimal escape force of 2.3 pN at a distance equal to the bead radius, that is R = 0.5 µm (*Ashkin, 1992*). This value lies in the lower range of forces measured by optical tweezers at the growth cone periphery (*Thoumine et al., 2006*). As a control for the specificity of microsphere binding to endogenous N-cadherin at the growth cone surface, we compared the movement of beads coated with Ncad-Fc and Fc alone (*Figure 8A*). Fc-coated beads rarely escaped the optical trap within the 2 min recording, thereby exhibiting a greatly reduced escape probability (8% for Fc alone vs 70% for Ncad-Fc on control growth cones). Significantly, the velocity of Ncad-Fc beads after escape (V = 0.12 ± 0.01 µm/s, 50 beads) was very similar to the actin flow speed measured earlier by sptPALM on actin-mEos2, strengthening the idea of a direct transmembrane coupling. Ncad-Fc-coated beads placed on *Vangl2* cKO growth cones escaped the optical trap earlier than beads on control growth cones after 1 min, but also traveled longer distances (*Figure 8A,B,D*). Importantly, the absence of Vangl2 did not affect the bead velocity after escaping the trap (*Figure 8A,B,C*), indicating that the actin flow is altered only when N-cadherin adhesions are engaged. Finally, we measured the escape probability of beads as described previously (*Bard et al., 2008*), and found that this index is significantly increased (+20%) when beads are placed on top of V*angl2* cKO growth cones (*Figure 8E*). These data support the model that the absence of Vangl2 increases the coupling efficiency between N-cadherin adhesions and the actin flow, thereby influencing the speed of actin flow and, ultimately, the displacement of growth cones.

## Discussion

Our study shows that the absence of Vangl2 in young neurons increases the migration rate of the neuronal growth cone, due to a reduced turnover of N-cadherin adhesions resulting in an increased mechanical connection to the actin retrograde flow. Collectively, these observations support a model in which endogenous Vangl2 acts as a negative regulator of a molecular clutch between N-cadherin adhesions and F-actin in growth cones (Model, *Figure 9*).

It is well established that the inactivation of core PCP genes *Celsr3* and *Fzd3* leads to severe defects in multiple major axon tracts of the central nervous system (*Feng et al., 2012*; *Hamblet et al., 2002*; *Hua et al., 2014*; *Tissir and Goffinet, 2013*). It was further shown that neither *Celsr3* nor *Fzd3* deletion affect axonal outgrowth but rather disrupt the ability of the growth cones to respond to guidance cues (*Chai et al., 2014*; *Hamblet et al., 2002*; *Hua et al., 2014*). In contrast, we show here that the inactivation of *Vangl2* affects neural outgrowth, and more specifically, that the outgrowth of *Vangl2* cKO hippocampal neurons is sensitive to N-cadherin-mediated adhesion. These results suggest that core PCP genes do not use a conserved signaling cassette in young neurons as they do in epithelial cells, at least for axon guidance, and can activate specific signaling pathways independent from each other (*Guo et al., 2004*; *Klein and Mlodzik, 2004*).

Our in vitro data, in a very controlled environment and in the absence of any other interfering bias, allowed the identification of a clear cell autonomous deficit due to the absence of Vangl2 in the neuron. For a growth cone to advance, it must not only be able to stick to the substrate but also be able to dynamically disassemble and reassemble adhesion complexes quickly. In fact,

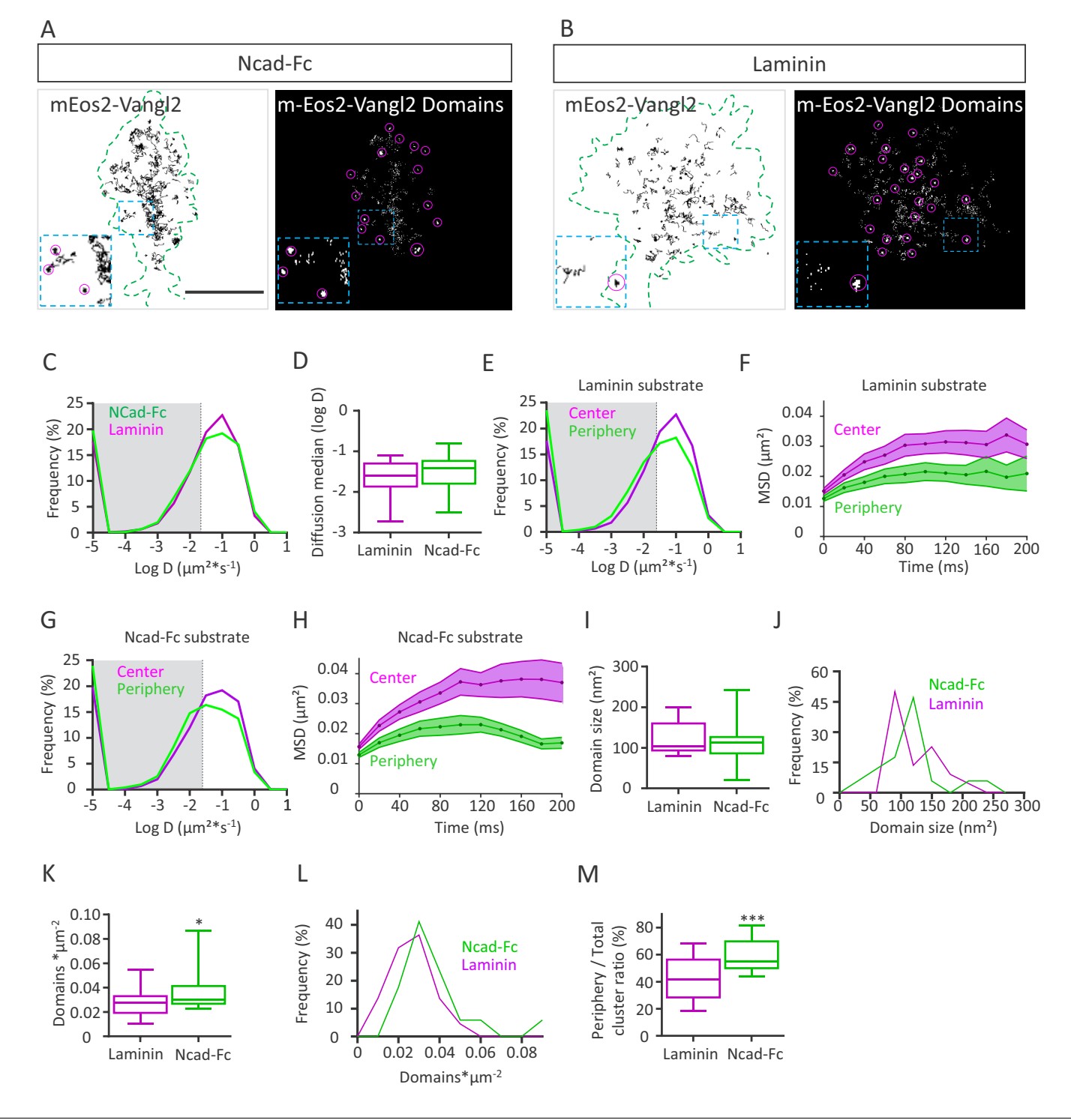

**Figure 7.** N-cadherin substrate affects Vangl2 distribution. (**A, B**) Representative images of growth cones from rat neuronal growth cones (depicted in green) expressing GFP and mEos2-Vangl2 on Ncad-Fc-coated (**A**) or laminin-coated substrates (**B**). The images on the left show the trajectories of mEos2-Vangl2 molecules (black); images on the right show the corresponding super-resolution projections of all the single molecules detected. Scale bar, 2.5 μm. (**C**) Frequency distribution of diffusion coefficients of mEos2-Vangl2 on Ncad-Fc (green) or laminin substrates (magenta) on a logarithmic scale. (**D**) Median diffusion on both substrates. (**E**) Frequency distribution of diffusion coefficients of mEos2-Vangl2 on laminin on a logarithmic scale according to their localization at the periphery (magenta) or the center (green) of the growth cone. (**F**) Plot of the MSD of mEos2-Vangl2 molecules in growth cones plated on laminin substrate as a function of time. (**G**) Frequency distribution of diffusion coefficients of mEos2-Vangl2 on Ncad-Fc on a

*Figure 7 continued on next page*

*Figure 7 continued*

logarithmic scale according to their localization at the periphery (magenta) or the center (green) of the growth cone. (H) Plot of the MSD of mEos2-Vangl2 particles in growth cones plated on Ncad-Fc substrates as a function of time. n = 7707–11906 trajectories, 19–22 neurons. (I) Quantification of the average domain size of mEos2-Vangl2 in global growth cones. (J) Frequency distribution of mEos2-Vangl2 confinement domain size on Ncad-Fc or laminin substrate. (K) mEos2-Vangl2 domain density (domains*$\mu m^{-2}$) in global growth cones on Ncad-Fc and laminin substrates. (L) Frequency distribution of mEos2-Vangl2 domain density on Ncad-Fc or laminin substrate. (M) Ratio of mEos-Vangl2 domain density at the periphery versus the entire area of the growth cone on Ncad-Fc and laminin substrates. n = 472–962 domains, 17–21 neurons. Data are presented as box-and-whisker plots (min/max) from three experiments; *p<0.05 and ***p<0.001 by the Mann-Whitney test (K) or Student's t-test (M).

The online version of this article includes the following figure supplement(s) for figure 7:

**Figure supplement 1.** Frequency and cumulative distribution of mEos2-Vangl2 molecules in growth cones.

comparative studies have shown that neuronal growth cones generate significantly lower traction forces than non-neuronal cell types (*Amin et al., 2013*; *Moore et al., 2010*) but also that hippocampal neurons do not adhere to the substrate as strongly as other neuronal cell types (*Koch et al., 2012*). One important difference between the cells with strong, stable, and mature focal adhesions and neuronal growth cones is that in the latter, substrate coupling and adhesions are transient and rapidly evolving (*Koch et al., 2012*). Our data strongly suggest that Vangl2 participates in this dynamic junctional remodeling role, allowing the hippocampal neurons to adapt to the environment encountered while navigating. In a study on commissural motor neurons, *Shafer* et al. showed that downregulation of *Vangl2* levels via shRNAs prevented Wnt5a-stimulated axonal outgrowth (*Shafer et al., 2011*). In contrast, our results in hippocampal neurons show that neurite outgrowth is increased in the absence of Vangl2 but that Wnt5a-stimuated axonal outgrowth is not affected in *Vangl2* cKO neurons. We may envision that hippocampal and motor neurons respond differently to defects in Vangl2 signaling. This has been illustrated for *Fzd3* mutants both in terms of axonal

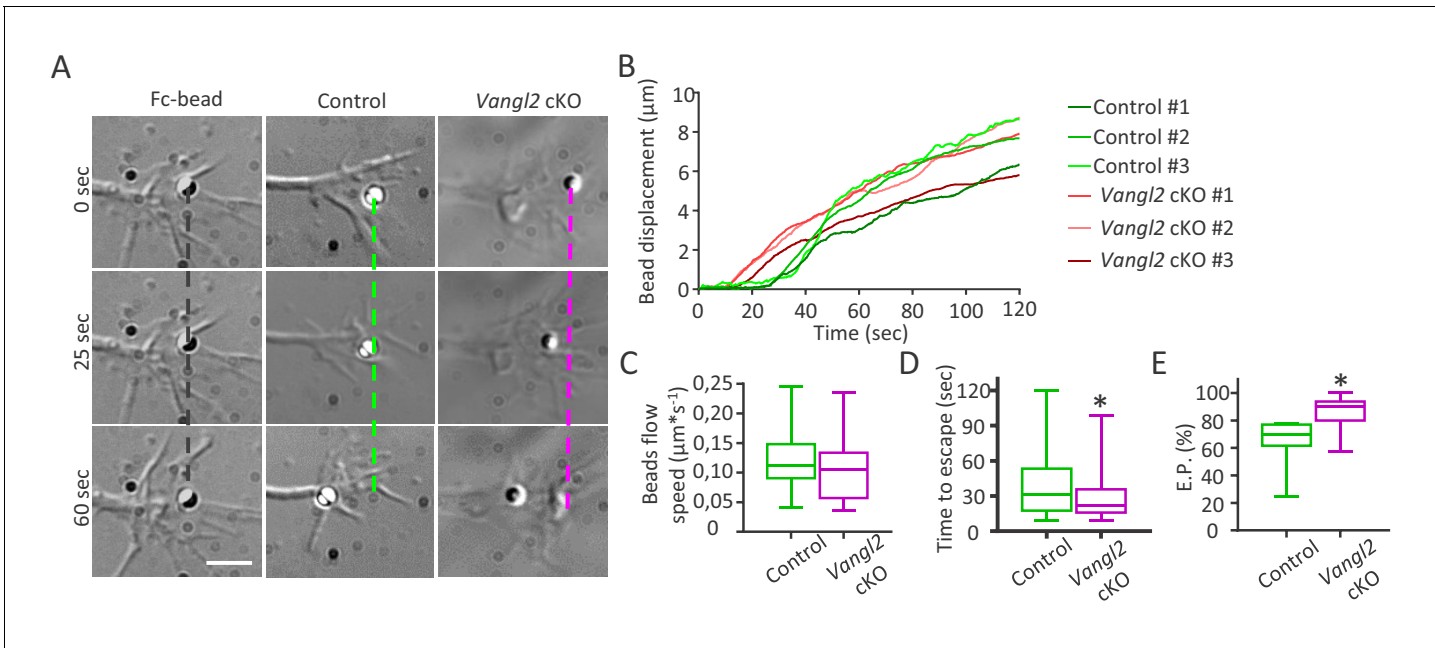

**Figure 8.** Vangl2 regulates actin-N-cadherin coupling at the growth cone surface. (A) Latex microspheres (1 µm) coated with either Fc fragment alone (left panel) or Ncad-Fc were placed and restrained for 2 min at the periphery of growth cones using optical tweezers for control or *Vangl2* cKO neurons. Scale bar, 2.5 µm. (B) Graph showing the displacement versus time for representative Ncad-Fc coated beads on growth from *Vangl2* cKO or control neurons. (C) Bead velocity calculated from the initial slope of the displacement versus time in the peripheral region (from 1 to 4 µm), after the beads have escaped the optical trap. (D) Quantification of the average time needed for the beads to escape from the center of the trap when placed on control or *Vangl2* cKO growth cones. (E) Quantification of the escape probability (E.P., calculated over the first 60 s of the recordings) for beads placed on either control or *Vangl2* cKO growth cones. n = 54–74 beads (12 for Fc-beads). Data are presented as box-and-whisker plots (min/max) from at least six different mice per genotype and two independent experiments. *p<0.05 by the Mann-Whitney test (D, E).

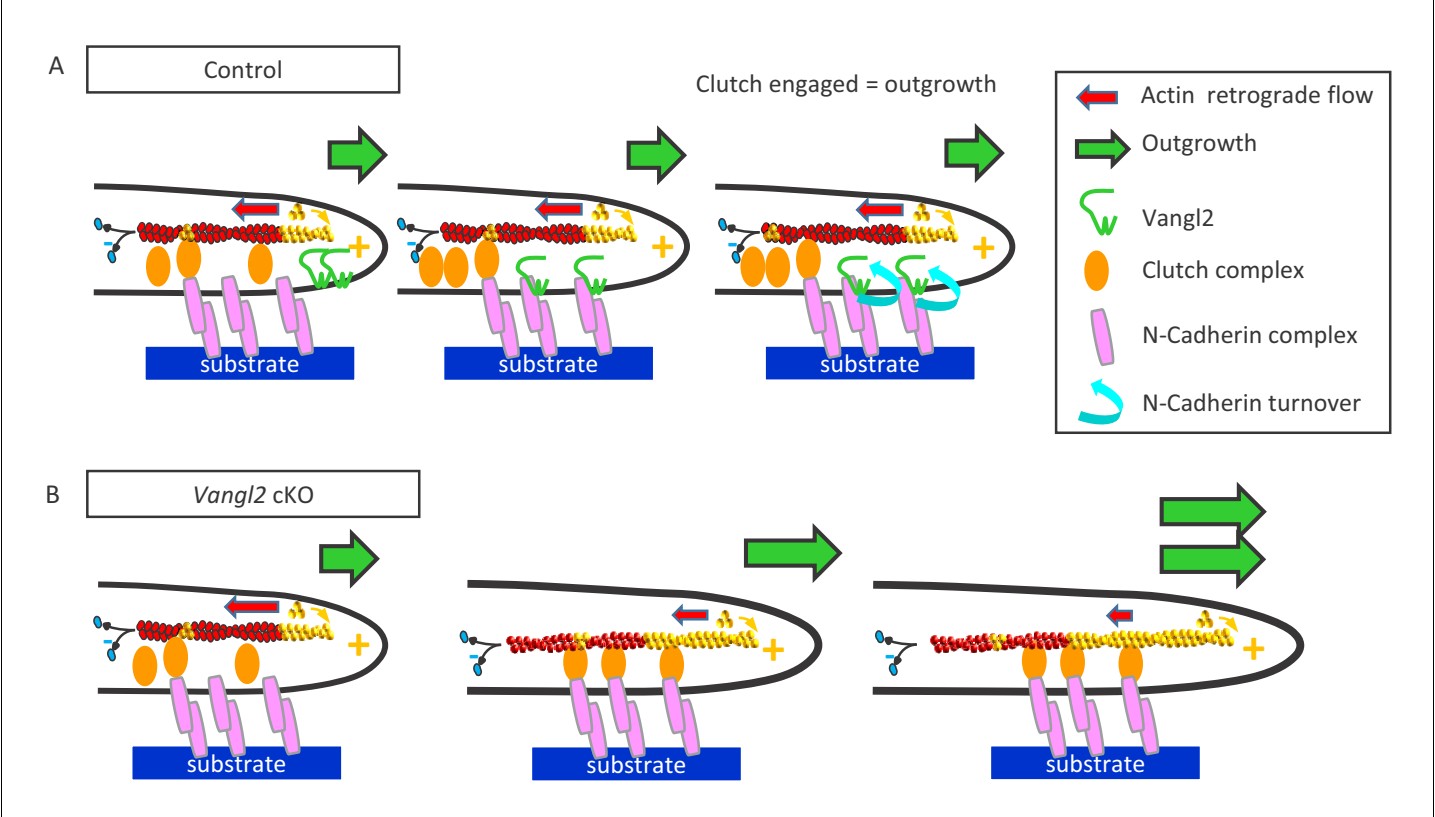

**Figure 9.** Vangl2 inhibits neuronal outgrowth by regulating N-cadherin engagement in the clutch mechanism. (A) The retrograde actin flow is generated by constant polymerization (at the tip of filopodia/lamellipodia) and depolymerization (minus end) of actin monomers along a filament (F-actin), and participates to the extension and retraction of filopodia/lamellipodia. When permissive substrates such as N-cadherin or laminin are present in the environment they bind to corresponding cell surface receptors at the plasma membrane, which recruit specific adhesion complexes, creating a molecular link between the adhesion proteins and F-actin filaments within the growth cone. This 'molecular clutch' leads to the local engagement of the actin flow, causing a decrease in its speed and a proportional increase in neuronal outgrowth, due to the ongoing actin polymerization. The presence of Vangl2 at the membrane participates in the regulation of this outgrowth by controlling the number of N-cadherin molecules engaged in the clutch. If no permissive substrate is present, the link is absent, and the outgrowth is slow, while the retrograde flow is fast. (B) The absence of Vangl2 leads to an increased presence/stabilization of N-cadherin at the plasma membrane, and the engagement of more N-cadherin molecules which in turn increase the molecular clutch strength. As a result, a decreased actin retrograde flow speed and an increased neuronal outgrowth are observed.

growth and guidance, possibly because of distinct responses to a specific diffusible cue (secreted or extracellular matrix - ECM) encountered as the axon grows (*Hua et al., 2014*).

At a molecular level, it is tempting to conclude that Vangl2 is a regulator of endocytosis in the context of junctional remodeling during tissue growth, including cell movement. Vangl2 has been shown to interact with a large variety of proteins, many of which still await characterization (*Bailly et al., 2018*). Within these interactors, there is now accumulating evidence in various species and tissues that E-cadherin and N-cadherin turnover is modulated by Vangl2 and its orthologues, through direct or indirect interaction (*Classen et al., 2005*; *Warrington et al., 2013*; *Nagaoka et al., 2014a*; *Nagaoka et al., 2014b*). Vangl2 has been shown to regulate the endocytosis of other molecules including Frizzled 3 (*Shafer et al., 2011*), nephrin (*Babayeva et al., 2013*) or the matrix metalloproteinase MMP14 (*Williams et al., 2012*). Therefore, Vangl2-mediated membrane protein endocytosis

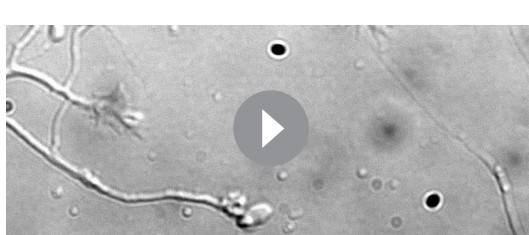

**Video 1.** Time lapse video with Differential Interference Contrast (DIC) microscopy of a control growth cone growing on NCad-Fc substrate for 45 min.
https://elifesciences.org/articles/51822#video1

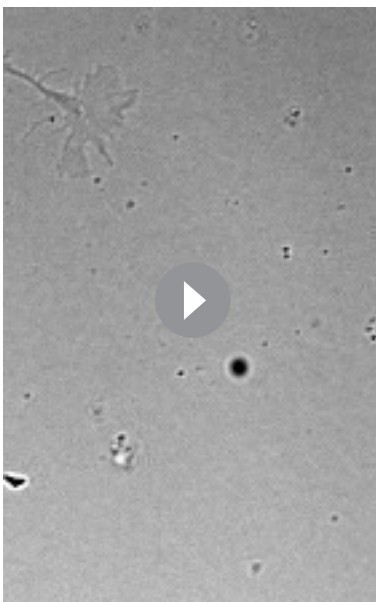

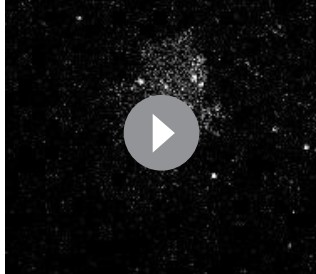

**Video 3.** sptPALM-TIRF acquisition of actin-mEos2 molecules in a control growth cone plated on NCad-Fc substrate.

https://elifesciences.org/articles/51822#video3

**Video 2.** Time lapse video with DIC microscopy of a *Vangl2* cKO growth cone growing on NCad-Fc substrate for 45 min.

https://elifesciences.org/articles/51822#video2

might represent an evolutionarily conserved mechanism by which tissue morphogenesis (including neuronal tissue) is controlled in both vertebrates and invertebrates.

In vivo *we* could imagine that an increase in axonal growth would lead the axons to reach the midline too early, before local cues necessary for their crossing are expressed, and as a result the axons would use alternate trajectories that are permissive to their passage. But the in vivo phenotype observed in the Emx1-Cre*Vangl2* cKO is more realistically the result of a combination of the disruption of both autonomous and non-autonomous Vangl2-dependant mechanisms. Vangl2 is broadly expressed in the cortex and hippocampus early in development and notably at the time of intense neurogenesis (*Tissir and Goffinet, 2006*, *this manuscript*) and its deletion could affect various mechanisms, in neuronal and non-neuronal cells. The environment explored by the growth cone as it moves might be affected by *Vangl2* deletion, in terms for example of guideposts location/maturity, secretion of diffusible molecules, fasciculation or gene expression, including ECM proteins expression levels. Our data on the redistribution of Vangl2 at the periphery of the growth cone on the N-cadherin substrate suggest that the composition of the ECM could reciprocally modulate the distribution of Vangl2 molecules in the growth cone and then modulate its behavior. It seems reasonable to think that this effect could be mediated

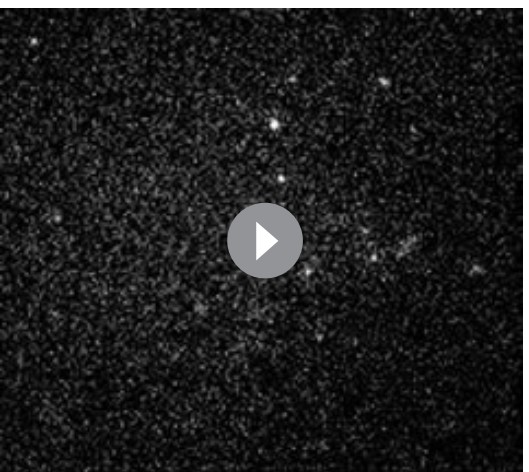

**Video 4.** sptPALM-TIRF acquisition of actin-mEos2 molecules in a *Vangl2* cKO growth cone plated on NCad-Fc substrate.

https://elifesciences.org/articles/51822#video4

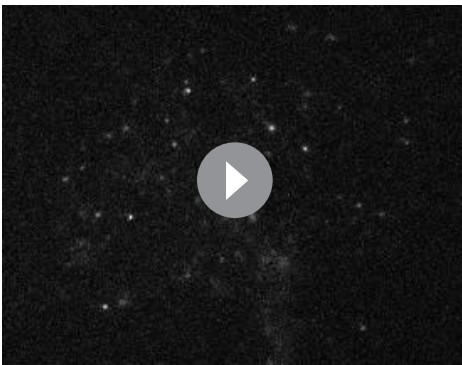

**Video 5.** sptPALM-TIRF acquisition of actin-mEos2 molecules in a control growth cone plated on PLL substrate.

https://elifesciences.org/articles/51822#video5

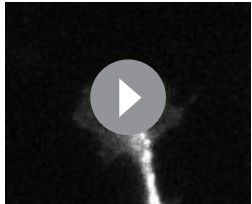

**Video 6.** Time lapse acquisition of EB3-GFP comets in a control growth cone plated on NCad-Fc substrate.
https://elifesciences.org/articles/51822#video6

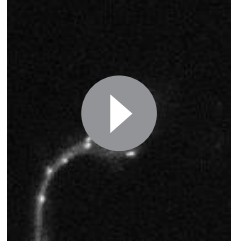

**Video 7.** Time lapse acquisition of EB3-GFP comets particles in a *Vangl2* cKO growth cone plated on NCad-Fc substrate.
https://elifesciences.org/articles/51822#video7

via homophilic N-cadherin interactions in trans transmitted via direct or indirect interaction with Vangl2. In vivo, we could imagine that guidance is regulated by the interplay between Vangl2 and the ratio of N-cadherin and laminin (ECM composition) present along the growth cone path as it grows. When the growth cone encounters an N-cadherin-rich environment, Vangl2 could control the number of pauses (for example), allowing better exploration, and possibly influencing the orientation of the growth cone. But if this is the case, most of the forebrain neurons appear to be able to compensate for the absence of such a Vangl2 control of guidance in vivo. Thus, we believe that the clear differences in neuronal behavior and in the corresponding in vivo phenotypes, similar for Fzd3 and Celsr3 but different from Vangl2, suggest that the N-cadherin/Vangl2 relationship identified here in hippocampal neurons mostly pertains to outgrowth regulation (*Chai et al., 2014*; *Hua et al., 2013*; *Hua et al., 2014*; *Qu et al., 2014*; *Zhou et al., 2008*). Though mechanistically different, these results expand a growing body of literature highlighting some links between Vangl2 and components of the ECM during cell migration, notably during convergent extension in the zebrafish (*Dohn et al., 2013*; *Jessen and Jessen, 2017*; *Love et al., 2018*).

In the Emx1-Cre\**Vangl2* cKO conditional model used in this study, the agenesis of the cc is restricted to the posterior portion which is typical of a developmental deficit, as formation of the cc starts from the anterior part and progresses to the rear. In this same caudal region, it is suggested that cc crossing may be facilitated by hc (*Livy and Wahlsten, 1997*; *Ozaki and Wahlsten, 1992*;

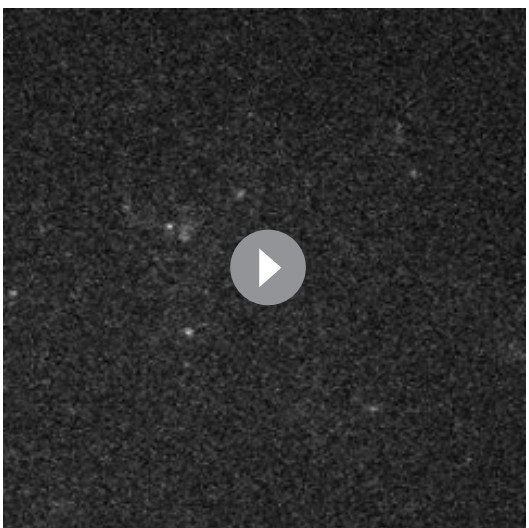

**Video 8.** sptPALM-TIRF acquisition of Ncad-mEos2 molecules in a control growth cone plated on NCad-Fc substrate.
https://elifesciences.org/articles/51822#video8

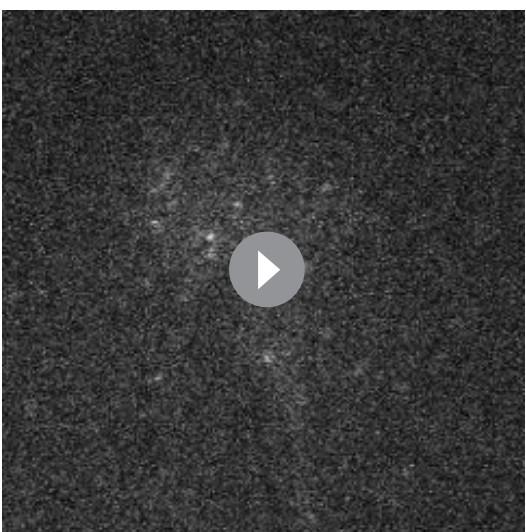

**Video 9.** sptPALM-TIRF acquisition of Ncad-mEos2 molecules in a *Vangl2* cKO growth cone plated on NCad-Fc substrate.
https://elifesciences.org/articles/51822#video9

*Wahlsten, 1981*). Although we cannot exclude that cortical neurons might also react abnormally to the absence of Vangl2, we propose that in the *Vangl2* cKO the cc deficit is, to a large extent, a secondary consequence of the faulty formation of the hc. In our mutant, the cc deficit is not associated with a significant disruption of upper-layer neurons or cortical thickness reduction, as observed in the Loop-tail mouse (*Lake and Sokol, 2009*). We believe this discrepancy can be partially explained by the absence of severe neural tube defects in the Emx1-Cre*Vangl2* cKO, or in a double *Vangl1*$^{-/-}$;Foxg1$^{Cre}$;*Vangl2*$^{f/f}$ compound cKO (*Qu et al., 2014*), as compared to mice with germline mutation/deletion such as the *Vangl2*$^{Lp/Lp}$ mutant or the null *Vangl2*$^{\Delta/\Delta}$ mutant (*Song et al., 2010*), which present with craniorachischisis. As such, some of the central nervous system defects seen in *Vangl2*$^{Lp/Lp}$ or in *Vangl2*$^{\Delta/\Delta}$ embryos brains could be a secondary consequence of the severe disruptions in tissue morphogenesis as observed in these mutants. Finally, a compounding factor in the *Vangl2*$^{Lp/Lp}$ embryo could be the existence of dominant-negative or gain-of-function properties of the Vangl2$^{Lp}$ protein, as shown in other systems or suggested in previous studies (*Qu et al., 2014*; *Song et al., 2010*; *Torban et al., 2004*; *Yin et al., 2012*). It is therefore probable that at least some of the severe phenotype in cortical thickness observed by Lake and Sokol are not entirely due to a disruption in the neurogenesis process. This emphasizes again the caution necessary when drawing mechanistic conclusions on the role of Vangl2 (or Vangl1) when using the *Loop-tail* model.

In summary, this work offers an original and dynamic view of the role of Vangl2 in neuronal cells and gives us insight into how a specific core PCP gene can affects the development and the morphogenesis of neuronal tissue. This is specifically important given the links between core PCP genes and neurodevelopmental or neurological pathologies like neural tube defects and epilepsy (*Sans et al., 2016*).

# Materials and methods

## Key resources table

| Reagent type (species) or resource | Designation | Source or reference | Identifiers | Additional information |
|---|---|---|---|---|
| Strain, strain background (*M. musculus*) | *Emx1-Cre* | Jackson Laboratory (Bar Harbor, ME). | Emx1$^{B6.129S2\text{-}Emx1tm1(cre)Krj/J}$ | RRID:IMSR_JAX:005628 |
| Strain, strain background (*M. musculus*) | *Vangl2* flox/flox | *Ramsbottom et al., 2014* | | |
| Recombinant DNA reagent | actin-mEos2 | *Garcia et al., 2015* | | |
| Recombinant DNA reagent | N-cadherin-mEos2 | *Garcia et al., 2015* | Uniprot : X07277 | |
| Recombinant DNA reagent | N-cadherin-GFP | *Thoumine et al., 2006* | Uniprot : X07277 | |
| Recombinant DNA reagent | EB3-eGFP | *Leterrier et al., 2011* | | |
| Recombinant DNA reagent | pEGFP-C3 | Clontech | | |
| Recombinant DNA reagent | mEos2-Vangl2 | This paper | Uniprot : Q91ZD4 | |
| Transfected construct | AAV9-hSyn1-GFP-2A-GFP-f | Given by Dr G Feng; Zhang et al J Neurosci. 2016; PMID: 26888934 | | Adeno-associated virus construct to transfect and express GFP. Titer: $2.4 \times 10^{13}$ gcp/ml |
| Other | Latex microspheres | Polysciences, | ref: 19404 | |
| Peptide, recombinant protein | ChromPure Human IgG, Fc fragment | Jackson Immuno Research | ref: 009-000-008 | 0.04 mg/mL |

*Continued on next page*

*Continued*

| Reagent type (species) or resource | Designation | Source or reference | Identifiers | Additional information |
|---|---|---|---|---|
| Peptide, recombinant protein | Recombinant human N-cadherin Fc chimera Protein, CF | R and D Systems | Ref 1388-NC-050 | 0.04 mg/mL |
| Antibody | AffiniPure Goat anti-human IgG, Fcγ Fragment specific | Jackson Immunoresearch | Ref 109-005-098 | 0.023 mg/mL |
| Antibody | Rabbit polyclonal anti-Vangl2 | *Montcouquiol et al., 2006* | | 1/500 |
| Antibody | guinea pig polyclonal anti-Doublecortin-X | Millipore | Ref AB2253 | 1/2000 |
| Antibody | Mouse monoclonal anti-β-catenin | BD Biosciences | clone 14 | 1/200 |
| Antibody | Mouse monoclonal anti-GAPDH | Millipore | Ref AB 2302 | 1/5000 |
| Antibody | Amersham ECL Mouse IgG, HRP-linked whole Ab (from sheep) | GE Healthcare UK | Ref NA931 | 1/5000 |
| Antibody | Amersham ECL Rabbit IgG, HRP-linked whole Ab (from donkey) | GE Healthcare UK | Ref NA934 | 1/5000 |
| Antibody | Rabbit polyclonal anti-pan actin | Cytoskeleton | Ref AANO1 | 1/500 |
| Antibody | Mouse monoclonal anti-N-cadherin | BD Biosciences | clone 32 | 1/200 |
| Antibody | Mouse monoclonal Anti-Neurofilament-H | Hybridoma Bank | 2H3 | 1/100 |
| Antibody | Rabbit polyclonal anti-Cux1 | Santa Cruz | Ref Sc13024 | 1/200 |
| Other | Fluoromyelin | Thermo Fisher Scientific | BrainStain Imaging kit, Ref B34650 | 1/300 |
| Other | Phalloidin Coumarin | Sigma | Ref P2495 | 1/500 |
| Software | MetaMorph | Molecular Devices | | |
| Algorithm | PALM Tracer | *Kechkar et al., 2013* | JB Sibarita | Custom made algorithm |
| Software, algorithm | Mathematica 4.1 version, Wolfram Research | *Czöndör et al., 2012*; *Garcia et al., 2015* | O Thoumine | Custom made algorithm |
| Software | Prism 6.0 | Graphpad software | | |
| Software | Hamamatsu NDP viewer | Hamamatsu | | |

## Transgenic mice

All procedures involving animals were done in accordance to the European Communities Council Directives (2010/63/EU) and the French National Committee (2013–118) recommendations. The French 'Ministere de l'Education Nationale, de l'Enseignement Superieur et de la Recherche' approved all experiments under the authorization n°APAFIS#1360–2015080317 20985 after agreement from the ethical committee of the University of Bordeaux. The Vangl2 flox transgenic is detailed in *Ramsbottom et al. (2014)*. Emx1[B6.129S2-Emx1tm1(cre)Krj/J] (*Emx1*-Cre) stocks were obtained from the Jackson Laboratory (Bar Harbor, ME). The *Vangl2* conditional knockout line (cKO) was generated by crossing *Vangl2* flox/flox animals with Emx1-Cre animals (*Vangl2* cKO), effectively deleting full-length Vangl2 in the telencephalon as early as E10.5. The recombination following the expression of Cre is predicted to produce a premature stop codon producing an 8 KDa protein, which lacks the four trans-membrane domains and C-terminal domain, resulting in an effective loss of function.

## Histology

For histology, brains were harvested from postnatal day 14 (P14) mice and fixed in Bouin's fixative (Electron Microscopy Sciences) overnight (ON), dehydrated in ethanol, paraffin-embedded, and coronal sections (20 µm) obtained, before being stained with hematoxylin and mounted with Entellan (Millipore). Brain sections were examined using Leica MZ-16 stereomicroscope using the Nano-Zoomer 2.0-HT slide scanner and analyzed with the Hamamatsu NDP viewer software (Hamamatsu).

## 3d volume reconstitution and surface rendering

Adult (10 week-old) controls and *Vangl2* cKOs mice were perfused transcardially with PB followed by 4% PFA in PBS; the brains were removed and postfixed in 4% PFA for 48 hr at 4˚C before sectioning. All coronal sections were stained with cresyl violet, scanned with Hamamatsu NANOZOOMER 2.0. compiled into a stack file after being aligned using ImageJ software. Stacks were projected and compiled into a single image with Imaris Scientific 3D/4D image processing and analysis software. Manual segmentation was performed for 3D reconstruction of total corpus callosum and hippocampal regions (Hippocampal formation and dorsal commissure). The thickness of the somatosensory cortex was measured on the same coronal sections stained with cresyl violet. We selected 3 to 9 non-consecutive slides at the level of the dorsal hippocampus, and measured cortex thickness on each side at a distance of 2 mm from the midline.

## Immunohistochemistry

For neurofilament staining, embryonic day 12.5 (E12) to E14.5 embryos were fixed by immersion ON at 4˚C in phosphate buffer (PB) with 4% paraformaldehyde (PFA). 15 µm cryosections oriented coronally were processed for immunostaining with anti-neurofilament (NF) mouse monoclonal antibody 2H3 (Developmental Studies Hybridoma Bank, Iowa City,IA) using a 1:50 dilution in PBS with 0.3% Triton X-100 ON at 4˚C. For P21 and P31 brains, the animals were perfused transcardially with PB followed by 4% PFA, and the brains were post-fixed in 4% PFA in PB for 24 hr at 4˚C. Coronal or horizontal vibratome sections (40 µm) were obtained and processed for NF staining or fluoromyelin labeling (BrainStain Imaging kit, B34650, Thermo Fisher Scientific, 1:300). For Cux1 staining, brains from P0 were fixed ON 4˚C in PB with 4% PFA, and 15 µm cryosections oriented coronally were processed for immunostaining with anti-Cux1 antibody (Santa Cruz, sc130124, rabbit polyclonal) ON at 4˚C using a 1:200 dilution in PBS with 0.3% Triton X-100.

For all the different developmental and postnatal stages, images were taken with the same exposure time for all genotypes on a Zeiss AxioImager Z1 and an AxioCam MRm. Analysis of Cux1 positive cells were performed in somatosensory cortical regions. In those regions, subregions of 100 µm of width were used in order to do cell counts along the entire cortical wall.

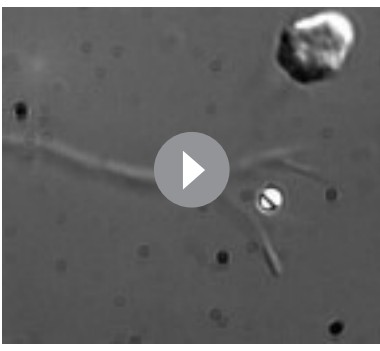

**Video 10.** Optical trapping and escaping of a Ncad-Fc coated bead placed on the dorsal surface of a control growth cone.
https://elifesciences.org/articles/51822#video10

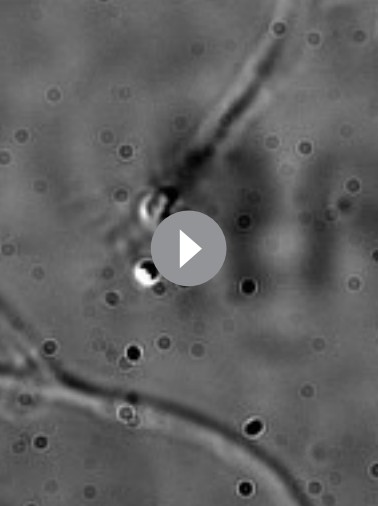

**Video 11.** Optical trapping and escaping of a Ncad-Fc coated bead placed on the dorsal surface of a *Vangl2* cKO growth cone.
https://elifesciences.org/articles/51822#video11

## Viral stereotaxic injections in hippocampus

Twenty-week-old controls and cKOs mice received unilateral stereotaxic microinjections of a AAV (0.5 µl at 300 nl per min) expressing GFP under the control of the promoter hSyn1 (AAV9-hSyn1-GFP-2A-GFP-f, titer $2.4 \times 10^{13}$ GC/ml, *Zhang et al., 2016*) in the CA3 region of hippocampus (anteroposterior [Y] −1.75 mm from Bregma, mediolateral [X] −1.85 mm, dorsoventral [Z] −2 mm). Four weeks after surgery, the animals were perfused transcardially with PB followed by 4% PFA in PBS; the brains were removed and postfixed in 4% PFA in PBS for 24 hr at 4°C before sectioning, staining with DAPI and confocal analysis.

## Neuronal cell culture, immunofluorescence and cDNA constructs

For hippocampal neurons culture, hippocampi were harvested from P0/P1 *Vangl2* cKO and control littermates in an HBSS solution, enzymatically and mechanically dissociated (*Mauriac et al., 2017*). The hippocampi of each pup were cultured individually and blind of the genotype. After 14 min incubation at 37°C in trypsin, the hippocampi were rinsed with a Plating Medium (PM) that contains neurobasal medium, B27 supplement (50X, Gibco), 2 mM Glutamine, 0.3% Glucose, 37.5 mM NaCl and 5% of Fetal Bovine Serum (FBS). The hippocampi were dissociated by gentle up and down movements through a Pasteur pipette, and then centrifuged at 1 rpm, 5 min at room temperature (RT) after counting. The neurons were further diluted in 1 ml of PM and plated on the previously prepared coverslips. Two hours after plating, the PM was replaced by a similar medium without serum.

For experiments on N-cadherin substrate, we used a protocol similar to *Garcia et al. (2015)*. Regular 18 mm glass coverslips were incubated for 2 hr at 37°C with 1 mg/ml poly-L-lysine (PLL) in borate buffer, rinsed with $H_2O$, then incubated 2 hr at 37°C with 4 µg/cover of AffiniPure goat anti-human IgG, Fc$_\gamma$ Fragment specific, (109-005-098, Jackson Immunoresearch) in 0.2 M boric acid (pH 8.5), before another incubation ON at 4°C with 0.2 µg/cover of Recombinant human N-cadherin Fc chimera protein (Ref 1388-NC-050, R and D Systems) (stp-PALM of actin-mEos2, FRAP, EB3-GFP time lapse) and 0.6 µg/cover (neuronal outgrowth with DIC time-lapse, stp-PALM of N-cadherin-mEos2, mEos2-Vangl2). Before use, the coverslips were rinsed again with boric acid (0.2 M, pH 8.4). For experiments on laminin substrate, coverslips were coated with 1 mg/ml PLL 2 hr at 37°C followed by 5 µg/cover laminin (Ref L2020, Sigma-Aldrich) ON at 4°C. Finally, for experiments with PLL substrate, coverslips were coated with 10 ug/ml or 1 mg/ml PLL for 2 hr at 37°C.

For Vangl2 immunocytochemistry, approximately 200,000 cells were plated on coverslips treated with 0.1 mg/ml of poly-lysine followed by 2 hr incubation with 1 µg/ml of laminin. At DIV2, neurons were fixed for 10 min with 4% PFA in PB/4% sucrose at RT, then pre-incubated 30 min in permeabilization buffer (PBS, 10% Normal Goat Serum, 0.1% Triton X-100). Cells were then incubated at RT for 1 hr with rabbit anti-Vangl2 (*Montcouquiol et al., 2006*), guinea pig polyclonal anti-Doublecortin-X (AB2253, Millipore) and Phalloidin Coumarin (P2495, Sigma). Coverslips were then mounted with Prolong Gold antifade reagent (P36930, THERMOFISHER). Fluorescent images of the neurons were obtained using an epifluorescence microscope (Zeiss Axio Imager Z1), Axiovison software for image acquisition and further processed with Adobe Photoshop or ImageJ. For the Shape Index (S.I.) analysis we outlined the growth cone (including lamellipodia) of DIV2 neurons grown on laminin substrate using a Phalloidin labeling and measurements of area (A) and perimeter (P) were extracted (Metamorph). The S.I. is calculated by dividing the A ($4\pi A$) by $P^2$, resulting in values close to 0 for highly filopodial growth cones and close to one for very lamellipodial ones.

*DsRed-Vangl2* was generated by inserting mouse CDS Vangl2 in a commercially available *pDsRed-C1* vector. *Actin-mEos2*, *N-cadherin-mEos2* and *N-cadherin-GFP* were previously described (*Bard et al., 2008*; *Garcia et al., 2015*). *Vangl2-mEos2* was generated by replacing actin in the *Actin-mEos2* construct (BglII and BamHI ligation).

## Video-microscopy

For video-microscopy experiments (neuronal outgrowth), hippocampal neurons were plated at a density of 50,000 cells per coverslip, on different substrates as described. At DIV2, neurons were bathed in with 1 ml of Tyrode solution (120 mM NaCl, 5 mM KCl, 2 mM MgCl2, 2 mM CaCl2, 25 mM 4-(2-hydroxyethyl)−1-piperazineethanesulfonic acid, and 30 mM D-glucose, pH 7.4), and observed on Nikon Eclipse TiE microscope equipped with autofocus and motorized 2D stage. Temperature was maintained at 37°C. We used an oil 63x/1.40 N.A objective and differential interference

contrast illumination (DIC). Images were acquired every 1 min for 45 min using an Orca Flash 4.0 camera (Hamamatsu Photonics, Hamamatsu City, Japan) driven by MetaMorph software (Molecular Devices, Sunnyvale, USA). Multi-position recordings were performed with the Multi-Dimensional Acquisition Module. Quantification of neuron growth cone speed was performed using the ImageJ software plugin 'manual tracking'. The global growth cone velocity was calculated by dividing the total distance travelled for each growth cone (measured with Manual Tracking mode) by the total time of acquisition (45 min). Quantification of the number of pauses was performed for growth cones that presented a velocity close to the average value for each condition (mean ± SEM). For EB3-eGFP tracking, 500,000 hippocampal neurons were nucleofected (4D-Nucleofector, Lonza) after dissociation with 1 µg EB3-eGFP and then distributed in two different wells. Images were acquired on a Nikon Ti Eclipse (Nikon France S.A.S., Champigny-sur-Marne, France). Sequences of 360 images were acquired using 100 ms exposure time and at 2 Hz frequency (1 image every 0.5 s) in order to visualize long trajectories reflecting the slow movements of tubulin filaments.

## SptPALM-TIRF

To track individual mEos2-tagged actin, N-cadherin or Vangl2 molecules, we used live single particle tracking combined to PhotoActivation Localization Microscopy (sptPALM). For actin-mEos2 experiments, 250,000 hippocampal neurons from control and V*angl2* cKO mice were nucleofected with 1 µg *pEGFP-C3* and 4 µg of *actin-mEos2* and plated on N-cadherin-Fc-coated coverslips (or PLL only for control experiments with neurons from control mice) at a density of 100,000 cells per coverslip. For N-cadherin-mEos2 experiments, 250,000 hippocampal neurons from control and V*angl2* cKO mice were nucleofected with 1 µg *pEGFP-C3* and 4 µg of *N-cadherin-mEos2* and plated on N-cadherin-Fc-coated coverslips at a density of 250,000 cells per coverslip. For mEos2-Vangl2 experiments, 250,000 rat hippocampal neurons were transfected with 1 µg *pEGFP-C3* and 4 µg of *mEos2-Vangl2* and plated on N-cadherin-Fc- or laminin-coated coverslips at a density of 250,000 cells per coverslip. Images were acquired on a Nikon Ti Eclipse (Nikon France S.A.S., Champigny-sur-Marne, France) with a TIRF arm and using an objective Apo TIRF 100 X oil N.A. 1.49 and an EMCCD camera (Evolve, Photometrics, Tucson, USA). The microscope is linked through an optical fiber to a four-color laser bench (405 nm, 488 nm, 561 nm, and 642 nm; 100 mW each). Photoactivation was done at 405 nm, while the images were taken using the 561 nm laser. Illumination was done under Total Internal Reflection Fluorescence (TIRF) to focus on molecules localized in a narrow optical section (100 nm) close to the substrate. For actin-mEos2, sequences of 360 images were acquired using 250 ms exposure time and at 2 Hz frequency (1 images every 0.5 s). This ensures that fast-diffusing actin monomers which contribute to a blur in the images are eliminated from subsequent analysis and that only slowly moving actin molecules incorporated in actin filaments are retained. Our sptPALM experiments were performed on genetically modified mice while our methodological model study used rat growth cones (*Garcia et al., 2015*). We noted numerous quantitative and qualitative differences that we attribute mostly to the difference between species. Notably, we had to be more rigorous in defining what we considered to be the peripheral region of the growth cone in order to avoid contamination by actin molecules that are located more centrally, because the growth cone are smaller in mouse hippocampal neurons. Also, we observed a lower activation of actin-mEos2 molecules in mice growth cones compared to rat, with an overall lower number of trajectories. This is confirmed also when comparing the number of N-cadherin-mEos2 molecules activated in mice growth cones vs mEos2-Vangl2 molecules in rat growth cones. Indeed, for mEos2-Vangl2 we were able to photoactivate and detect molecules with only 20 ms exposure time during 4000 acquisitions while for N-cadherin-mEos2 molecules we had to increase the exposure time (50 ms) and no molecules were photoactivated/detected after 2000 acquisitions.

### Speed, trajectories and nanodomain analysis

Actin, Vangl2, and N-cadherin trajectories recorded by sptPALM were computed and analyzed using custom-made algorithms written as a MetaMorph plug-in (PALMTracer) as described in *Garcia et al. (2015)*. Single-molecule localization was performed using a wavelet-based algorithm, and trajectories were computed using a simulated annealing algorithm (*Izeddin et al., 2012*; *Kechkar et al., 2013*). The trajectory duration, which corresponds to the time during which single mEos2 fluorophores emit red light upon 561 nm laser illumination, follows an exponential distribution strongly

shifted to short values. Only trajectories longer than seven frames in regions of interest were considered for analysis, which yielded a median of around nine frames for actin-mEos2 (*Figure 4—figure supplement 1A*), 10 frames for N-cadherin-mEos2 (*Figure 6—figure supplement 1B*) and 11 frames for mEos2-Vangl2 on both substrates (*Figure 7—figure supplement 1A,B*). The proportion of trajectories with more than seven time points is 63% in control and 61% in *Vangl2* cKO growth cones respectively. For slow actin-mEos2 trajectories, the mean squared displacement (MSD) function was computed for each trajectory over time, and fit by the power law MSD = $4Dt^{\alpha}$, where t is the time, D is the diffusion coefficient, and the exponent α (values between 0 and 2) reflects the curvature of the MSD function and the type of movement (*Saxton, 1994*). For highly directed trajectories, the MSD is a quadratic function of time, thus α is close to 2. To classify the α exponent of actin-mEos2 trajectories, computer simulations were ran (see below in Materials and methods) in order to determine the thresholds for both confined and directed motions, which were set to α = 0.5 and 1.5, respectively. For mEos2-Vangl2 and N-cadherin-mEos2 trajectories, the Diffusion values calculated by the PALM-TRACER plugin were converted onto a logarithmic scale and then pooled for comparison. Trajectories were sorted into two groups: confined and mobile. Confined trajectories were defined as trajectories with D < 0.015 $\mu m^2 * s^{-1}$ (for N-cadherin-mEos2) or D < 0.022 $\mu m^2 * s^{-1}$ (for mEos2-Vangl2), corresponding to molecules that explored an area inferior to the one defined by the image spatial resolution ~ $(0.11\ \mu m)^2$ for N-cadherin-mEos2 or $(0.084\ \mu m)^2$ for mEos-Vangl2) during the time used to fit the initial slope of the MSD (*Kechkar et al., 2013*) (4points = 80 ms): For example, the N-cadherin-mEos2 $D_{threshold} = (0.11\ \mu m)^2/(4 \times 4 \times 0.02\ s)$~0.015 $\mu m^2 * s^{-1}$. Confined trajectories with negative MSD slopes were arbitrarily assigned D = $10^{-5}$ $\mu m^2 * s^{-1}$. To analyze confinement domains, we used PALM-TRACER plugin to create super-resolved images of the intensity of mEos2 molecules (Vangl2 and N-cadherin) to obtain a fluorescent intensity map of the proteins of interest. The domains were isolated from the surrounding signal using a fluorescence threshold. The fluorescence profile of each domain is then fitted in a 2D sigmoidal function to extract information about the size of the fluorescent area (x and y coordinates) as well as pixel intensity. Growth cone surface was calculated with Metamorph to yield the number of domains/$\mu m^2$. For EB3-eGFP, the velocity (V) of EB3-eGFP particles was calculated by fitting the individual MSDs with the function $4Dt + V^2t^2$. The lifetime (T) of each EB3-eGFP particle was also calculated by dividing the number of time points displayed by each particle by 2 (to take in account the 2 Hz frequency acquisition), giving then a quantification of the number of seconds in which the EB3-eGFP particle was tracked.

For kymograph analysis, we first made an image of the projection of the maximal fluorescence of each image composing the stack acquired with actin-mEos2 (sptPALM-TIRF). Metamorph was used to perform kymograph analysis at the periphery (~5 μm width) of the imaged growth cones.

## FRAP

FRAP experiments were done as described in *Garcia et al. (2015)*. Briefly, hippocampal neurons were nucleofected with 1 μg of C-terminally GFP-tagged WT chicken N-cadherin and plated on N-cadherin-Fc-coated coverslips at a density of 100,000 cells per coverslip. Experiments were run under the same setup used for sptPALM. The laser bench has a second optical fiber output connected to an illumination device containing galvanometric mirrors (iLas; Roper Scientific), allowing scanning of the beam onto regions of interest defined on the images. Switching between the two fibers for alternating between imaging and bleaching is performed in the millisecond range using a mirror. After acquiring a 10 s baseline at a frame rate of 1 Hz, rapid photobleaching of a peripheral region of a growth cone was achieved at higher laser power (6 mW output at the objective focal plane) during 200–300 ms. Fluorescence recovery was further recorded for 200 s at a frame rate of 0.5–2 Hz. The data were fitted by a diffusion/reaction model that uses three parameters: the fraction of free receptors $\varphi$, the diffusion rate taken from freely moving N-cadherin-GFP receptors $k_{diff}$, and the turnover rate of N-cadherin-GFP adhesive interactions $k_{reac}$ (*Figure 3C*).

## Microsphere preparation and optical tweezers experiments

Twenty microliters of 1 μm latex microspheres (19404 Polysciences) were rinsed three times with 0.2 M boric acid (pH 8.4) and then resuspended with 100 μl boric acid. To 50 μl of this suspension, we added 10 μl of AffiniPure goat anti-human Fc$_\gamma$ Fragment specific, (109-005-098, Jackson Immunoresearch) and then added boric acid to reach a final volume of 100 μl. After incubation ON at 4°C, we

rinsed the beads three times with the boric acid solution containing 1% BSA (A7638 Sigma). The final pellet was then suspended with 60 µl boric acid containing 1% BSA. This suspension was then split in half: one part was incubated for 3–4 hr at room temperature with 4 µg of recombinant human N-cadherin-Fc chimera protein (1388-NC-050, R and D Systems) while the other half was incubated with 4 µg of ChromPure Human IgG, Fc fragment (009-000-008 Jackson ImmunoResearch) alone (control experiment to test adhesion specificity). Beads were rinsed again 3 times and suspended in 50 µl of a solution similar to the one in which the neurons are cultivated (PM solution), containing also 1% BSA and 20 mM Hepes. The beads were kept on ice during the experiments.

An inverted microscope Nikon Ti Eclipse (Nikon France S.A.S., Champigny-sur-Marne, France) was fed through its epifluorescence port by a 1064 nm/1 W laser beam (LCS-DTL-322 LASER 2000, Pessac, France) reflected by dichroic mirror (FF700-SDi01, SemROCK). Microsphere trapping was achieved with a laser power of 300 mW at the back focal plane of a 100x/1.49 N.A. oil immersion objective. Microspheres captured using a motorized stage were placed at the periphery of growth cones and the trap was then left ON for the total duration of the experiment. The microsphere movement in DIC illumination was recorded for 2 min at 5 Hz. We quantified the stiffness of the optical trap by imposing fluid flow of known velocity (V) around trapped microspheres, and measuring the bead displacement $\delta$ at steady state. This was done by keeping the bead in the optical trap several microns above the substrate, and moving the motorized 2D stage at constant speed. The flow velocity around the bead is thus equal to the stage speed and was calculated from the displacements of beads stuck at the surface of the coverslip. The displacement of the bead from the trap center was calculated by image subtraction (with versus without force). This procedure was performed for varying 1064 nm laser power, to make sure we were in the regime of a linear dependence of trap force versus laser power. We then applied Stokes formula giving the drag force around a spherical object: $F = 6\pi R\eta V$, where R is the bead radius (R = 0.5 µm), $\eta = 10^{-3}$ Pa.s is the fluid viscosity, and V is the flow velocity (60 µm/s). This provides us with a lower estimate of the escaping force of N-cadherin coated beads on growth cones, which occurs when the bead reaches a displacement $\delta$ roughly equal to the bead radius (*Ashkin, 1992*). At maximal laser power, we measured a trap stiffness of 4.7 pN/µm and thus an escape force of 2.3 pN at an escape distance of R = 0.5 µm. The trajectories described by the beads were tracked using Metamorph. To analyze the time needed for the bead to escape the trap, only the beads that eventually escaped the trap were accounted for. The escape probability ($P_e$) was defined as the frequency of beads that escaped the optical trap (over the total amount of beads recorded) during the first half of the recordings (i.e. after 1 min).

## Western blotting

The cortex, hippocampi, cerebellum and spinal cord from at least three P0/1 or P14 (hippocampus only) mice were dissected, pooled and weighted. A buffer solution containing 50 mM Tris-HCl (Gibco) and a mix of proteases inhibitors (Roche) was added to the tissues according to their weight. Samples were homogenized and sonicated and protein concentration was evaluated through BCA analysis (BCA protein Assay Kit from Thermo Scientific). Protein concentration was adjusted for each sample and diluted with 4X SDS loading buffer containing 5% β-mercaptoethanol as previously described (*Mauriac et al., 2017*). Proteins were loaded on a 10% acrylamide gels and ran at 125 V for 1 hr 30. The proteins are then transferred for 2 hr 30 to a PVDF Immobilon membrane (Millipore) or to a Nitrocellulose membrane. The membranes are saturated with a TBS (25 mM Tris, pH7.5; 137 mM NaCl; KCl 3 mM; bidistilled water up to 2 L) solution containing 8% milk or 5% BSA for ON at 4° C. The following day membranes were rinsed and incubated with a rabbit polyclonal anti-Vangl2 antibody at 1/2000 (*Montcouquiol et al., 2006*) or a mouse monoclonal anti-N-cadherin at 1/2000 (clone 32, BD Biosciences) or a mouse monoclonal anti-β-catenin at 1/2000 (clone 14, BD Biosciences) and a mouse monoclonal anti-GAPDH at 1/5000 (AB 2302, Millipore) antibodies followed by secondary antibodies (donkey anti-rabbit or anti-mouse IgG conjugated to horseradish peroxidase, GE Healthcare UK) incubations. The membranes were processed with chemiluminescence (ECL, Thermo Scientific). The optic density (OD)*mm$^{-2}$ was analyzed for each detected band. These values were then normalized by taking into account the GADPH OD*mm$^{-2}$.

## Actin polymerization assay

The amount of F-actin and G-actin was evaluated according to the Cytoskeleton Actin Polymerization Assay Kit (BK037, Cytoskeleton) protocol. Cortical and hippocampal neurons from P0 controls and *Vangl2* cKO mice were dissociated and plated at a density of 200,000 cells per dish and shortly (1 min) stimulated with 20 mM KCl before harvesting at DIV3 in cold PBS to be pelleted and stored at −80°C. To separate F-actin (pellet) from G-actin (supernatant) a 1 hr centrifugation at 100.000 x g was performed on the lysate, then the pellet was re-suspended in a volume equivalent to the supernatant using F-actin depolymerizing buffer. Actin levels were quantified by immunoblot using a rabbit polyclonal anti-pan actin antibody (Cytoskeleton). For each condition we pooled three petri dishes. The expression levels were determined using a Bio-Rad Quantity One system with a GS800 calibrated densitometer, and represented as a percentage of control band intensity.

## Computer simulations of actin dynamics in virtual growth cones

The program to simulate actin dynamics in growth cones is based on previous work (Garcia et a., 2015). The algorithm to compute the successive positions of individual actin molecules was written with the Mathematica software (4.1 version, Wolfram Research). The geometry is chosen as a 2D conical structure, in accordance with the flat nature of growth cones, with radius 12 μm and half angle 45°, based on experimental measurements. A single molecule is characterized by its spatial coordinates (x, y) over time (t). The total length of the trajectories (160 s) is based on the durations of sptPALM recordings, and the time step of the simulations is set to $\Delta t = 50$ ms, corresponding to a typical camera frame rate (total of 3200 frames). The initial molecule position was defined in cylindrical coordinates as $r(0) = r_0$ and $\theta(0) = \theta_0$, chosen randomly between 0 and 12 μm and −45° and +45°, respectively. At each time step t, the (x, y) coordinates of the molecule are incremented by the distances ($\Delta x$, $\Delta y$), depending on whether the molecule is freely diffusing, bound to the retrograde flow, or interacting with transmembrane adhesions. In the diffusive regime, x(t) and y(t) coordinates are incremented by $(2D\Delta t)^{1/2} n_1$ and $(2D\Delta t)^{1/2} n_2$, respectively, where $n_1$ and $n_2$ were random numbers generated from a normal distribution, to account for the stochastic nature of diffusion. Free cytosolic actin monomers are considered to display fast random diffusion (D = 3 $\mu m^2 \ast s^{-1}$) (*McGrath et al., 1998*). When actin monomers reach a 0.5 μm distance to the leading edge of the growth cone, they are allowed to assemble into filaments with a polymerization rate 0.5 $s^{-1}$, and slowly move rearward with a velocity V = 0.05–0.15 $\mu m \ast s^{-1}$, as measured in sptPALM experiments. In that regime, the radial coordinate $r(t) = [x(t)^2 + y(t)^2]^{1/2}$ is then incremented by $-V\Delta t$, and the x and y rectangular coordinates are calculated as $x(t)=r(t) \cos[\theta(t)]$ and $y(t)=r(t) \sin[\theta(t)]$, plus slow diffusive terms $(2D'\Delta t)^{1/2} n_1$ and $(2D'\Delta t)^{1/2} n_2$, respectively, accounting for short range lateral motion (D'=0.003 $\mu m^2 \ast s^{-1}$) matching that measured experimentally. During rearward motion, actin filaments are allowed to transiently couple to substrate-immobilized adhesion complexes (in particular N-cadherin) at the substrate, with adjustable kinetic parameters $k_c$ and $k_u$ (in units of $s^{-1}$). Actin molecules can bind only if the probability of coupling in this time interval, $k_c\Delta t$, is greater than a random number n between 0 and 1 generated from a uniform distribution. The coupling rate $k_c$ is taken as the average frequency of the pauses made by actin molecules on N-cadherin coated substrates ($k_c = 0.2$ $s^{-1}$), and the uncoupling rate is varied between 0.001 $s^{-1}$ (reflecting high coupling), and 0.8 $s^{-1}$ (reflecting low coupling). These values represent the two extreme types of interactions observed experimentally on N-cadherin coated substrates (*Bard et al., 2008*). The ratio $k_c/k_u$ which we call 'coupling strength', represents the binding affinity between filamentous actin and immobilized adhesion complexes. In this trapping mode, bound actin molecules are set to diffuse with a low diffusion coefficient (D''=0.001 $\mu m^2 \, s^{-1}$), that is the positions are incremented by $(2D'\Delta t)^{1/2} n_1$ and $(2D'\Delta t)^{1/2} n_2$, respectively. Actin molecules stay bound until the probability for detachment $k_c \Delta t$, exceeds another random number n'. When actin molecules detach from N-cadherin adhesions, they are set to move rearward again, until they couple to N-cadherin again, or escape the flow area. Actin filaments are allowed to spontaneously depolymerize into monomers within the peripheral region at a rate of $k_d = 0.05$ $s^{-1}$, in agreement with measurements based on fluorescent speckle microscopy (*Van Goor et al., 2012*). Actin filaments reaching the base of the growth cone (4 μm lower region) without depolymerizing are forced to disassemble and randomly diffuse again. For simplicity, we do not allow free actin monomers to bind adhesion complexes.

The output text file (x, y, t) is processed in the custom-made program PALM-Tracer (*Kechkar et al., 2013*) run under the MetaMorph software (Molecular Devices), and allowing the construction of stacks of images representing actin motion over time, as described previously. A zoom factor determines the pixel size of the reconstructed image (typically 25 nm), similar to the one used in sptPALM analysis. Four hundred trajectories per condition were generated. To mimic fluorophore photophysics, molecules are allowed to be fluorescent for a limited number of planes along their trajectory, before photobleaching. Molecules are thus turned on at a random time (between 0 and 150 s), then switched off after a random time chosen in an exponential distribution reproducing the experimental trajectory lengths. To reproduce the experimental conditions of a 2 Hz time lapse movie with camera exposure time of 250 ms, simulated images are summed by groups of 5 ($5 \times 50$ ms=250 ms), then one image out of two is kept. This procedure ensures that freely diffusing actin monomers which contribute to a blur in the images are eliminated from subsequent analysis and that only slowly moving actin molecules incorporated in filaments are retained. This yields a stack of 320 frames, from which trajectories of more than seven frames are selected, as in the experiments. The mean squared displacement (MSD) is computed for each trajectory, and fitted with the power function MSD = $4Dt^{\alpha}$, where D is a diffusion coefficient and the exponent $\alpha$ (values between 0 and 2) reflects the type of trajectory (directed for $\alpha$ close to 2, Brownian for $\alpha$ around 1, and confined for $\alpha < 1$). The fact that $\alpha$ coefficient can be less than 1, thereby deviating from random diffusion, is a result of the slow sampling mode. For directed trajectories, fitting the MSD with the function $4D't + V^2t^2$ (*Qian et al., 1991*) gave a velocity component in good agreement with the one entered in the simulations, thus validating the overall approach.

## Code availability

The Mathematica code used for the modelisation model is available as a source code file (*Source code 1*) provided in the Supplementary Material.

## Quantification and statistical analysis

For each experiment, sample sizes were chosen on the basis of initial pilot experiments. Similar experiments reported in previous publications were further used to direct sample sizes. For every group of samples analyzed, the outliers (with a significance level of $\alpha$ = 0.05) were discarded (https://www.graphpad.com/quickcalcs/Grubbs1.cfm) *angl2* cKO neurons were subjected to a normality test. If samples passed this test, they were then tested with an unpaired t-test, and if not, they were submitted to a Mann-Whitney test. For the outgrowth experiments with Netrin-1 or Wnt5, data were subjected to an ANOVA 2way followed by a Tukey's Multiple comparisons test. Differences were considered significant for p-values$\leq$0.05;$\leq$0.01;$\leq$0.001.

## Acknowledgements

We thank Prof. G Feng (Massachusetts Institute of Technology, Cambridge, Massachusetts) for the kind gift of the *AAV9-hSyn-GFP-2A-GFP-f* virus. We thank C Medina, B Tessier and S Benquet for some primary rat cultures and maxi-preps, R Peyroutou for the cloning of *mEos2-Vangl2* and the help with actin polymerization assay. We thank A Chazeau, C Landmann and N Piguel for their technical assistance. We thank the animal and genotyping facilities members of the Neurocentre Magendie for technical assistance, notably M Jacquet, D Gonzales and co-workers. Part of the microscopy was done in the Bordeaux Imaging Center, a service unit of the CNRS-INSERM and Bordeaux University, member of the national infrastructure France BioImaging, where the help of C Poujol, S Marais and P Mascalchi is acknowledged. The 'Biochemistry and Biophysics Facility' of the Bordeaux Neurocampus is funded by the LABEX BRAIN (ANR-10-LABX-43). The help of Y Rufin is acknowledged. Finally, we thank Ana Dorrego-Rivas, Jerome Ezan and Claudia Racca for proof-reading the final manuscript.

## Additional information

### Funding

| Funder | Grant reference number | Author |
|---|---|---|
| Fondation pour la Recherche Médicale | Equipe FRM DEQ20160334899 | Mireille Montcouquiol |
| Agence Nationale de la Recherche | Grant ANR-12-BSV4-0016-01 | Nathalie Sans |
| Horizon 2020 Framework Programme | PhD fellowship ENC H2020 Neurasmus | Steve Dos-Santos Carvalho |
| Fondation pour la Recherche Médicale | Ph.D. fellowship fourth year | Steve Dos-Santos Carvalho |
| Conseil Régional d'Aquitaine | Grant Neurocampus Program | Nathalie Sans Mireille Montcouquiol |
| British Heart Foundation | Grant PG/11/76/29108 | Deborah J Henderson |
| Agence Nationale de la Recherche | Grant Labex BRAIN ANR-10-LABX-43 | Vincent Studer Olivier Thoumine Mireille Montcouquiol |

The funders had no role in study design, data collection and interpretation, or the decision to submit the work for publication.

### Author contributions

Steve Dos-Santos Carvalho, Data curation, Software, Formal analysis, Validation, Visualization, Methodology, Writing - original draft, Writing - review and editing, Cux1 thickness analysis, Mutant mouse culture, Neuron shape analysis, Video time lapse, sptPALM-TIRF, FRAP, Optical Tweezers, Neurofilament and Vangl2 immunofluorescence experiments on neurons, embryo and adult brains, Western Blotting; Maite M Moreau, Data curation, Formal analysis, Visualization, Methodology, Writing - review and editing, Virus injection, Cresyl violet staining, 3D rendering, Cortical thickness analysis; Yeri Esther Hien, Data curation, Formal analysis, Investigation, Methodology, Writing - review and editing, Video time lapse; Mikael Garcia, Supervision, Validation, Methodology; Nathalie Aubailly, Formal analysis, Methodology, Cresyl violet, Hematoxylin, Nanozoomer aquisition; Deborah J Henderson, Resources, Funding acquisition, Writing - review and editing; Vincent Studer, Data curation, Formal analysis, Funding acquisition, Methodology, Writing - review and editing; Nathalie Sans, Resources, Data curation, Formal analysis, Supervision, Funding acquisition, Methodology, Project administration, Writing - review and editing; Olivier Thoumine, Conceptualization, Resources, Software, Supervision, Funding acquisition, Methodology, Writing - review and editing; Mireille Montcouquiol, Conceptualization, Resources, Formal analysis, Supervision, Funding acquisition, Validation, Investigation, Methodology, Writing - original draft, Project administration, Writing - review and editing

### Author ORCIDs

Deborah J Henderson (id) https://orcid.org/0000-0002-2705-5998
Nathalie Sans (id) http://orcid.org/0000-0001-7658-2471
Olivier Thoumine (id) http://orcid.org/0000-0002-8041-1349
Mireille Montcouquiol (id) https://orcid.org/0000-0001-8739-6519

### Ethics

Animal experimentation: All procedures involving animals were done in accordance to the European Communities Council Directives (2010/63/EU) and the French National Committee (2013-118) recommendations. The French "Ministere de l'Education Nationale, de l'Enseignement Superieur et de la Recherche" approved all experiments under the authorization n°APAFIS#1360-2015080317 20985 after agreement from the ethical committee of the University of Bordeaux.

Decision letter and Author response
Decision letter https://doi.org/10.7554/eLife.51822.sa1
Author response https://doi.org/10.7554/eLife.51822.sa2

## Additional files

### Supplementary files

- Source code 1. Mathematica code used for the modelisation model.

- Transparent reporting form

### Data availability

All data generated or analysed during this study are included in the manuscript and supporting files. Examples of movies from which the data have been extracted are provided for: Figure 3A (Video 1 and 2); Figure 4A (Video 3, 4 and 5); Figure 4F (Video 6 and 7); Figure 6A (Video 8 and 9) and Figure 8A (Video 10 and 11).

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
