## [Decision Letter]

**Acceptance summary:**

Planar cell polarity (PCP) is important for morphogenesis and involves a polarized distribution of protein complexes that act on the cytoskeleton. In the context of axon outgrowth in mammals, the PCP components Fzd3 and Celsr3 play key functions in response to axon guidance cues and the formation of major axonal tracts, whereas the role of Vangl2 has remained unclear.

In this manuscript, the authors employed a sophisticated array of live imaging and molecular diffusion assays as well as a cortex-specific Vangl2 conditional knockout (cKO) mice to evaluate the regulation of actin treadmilling, N-cadherin dynamics, and growth cone behavior. These approaches yielded a compelling argument for a molecular mechanism by which Vangl2 regulates the growth cone velocity, without impairing the response to guidance cues, by acting as a molecular clutch that links actin treadmilling within the growth cone to the external growth substrate. Vangl2 is important for the retrograde actin flow and Ncadherin-coupling to the actin cytoskeleton; and in turn, the localization of Vangl2 in the growth cone relies on extracellular N-cadherin. They also provide evidence that the function of Vangl2 in actin and N-cadherin dynamics is required for the formation of the hippocampal commissure and corpus callosum.

**Decision letter after peer review:**

Thank you for submitting your article "Vangl2 acts at the interface between actin and N-cadherin to control mammalian neuronal outgrowth" for consideration by *eLife*. Your article has been reviewed by three peer reviewers, including Fadel Tissir as the Reviewing Editor and Reviewer #1, and the evaluation has been overseen by Anna Akhmanova as the Senior Editor.

The reviewers have discussed the reviews with one another and the Reviewing Editor has drafted this decision to help you prepare a revised submission.

Summary:

This is an impressive and technically elegant study explores in depth the underlying mechanisms by which the core PCP protein Vangl2 regulates the axonal outgrowth. It is extremely well quantified and analyzed and should be of great interest for the broad readership of *eLife*. The authors employed a sophisticated array of live imaging and molecular diffusion assays, and a vangl2 cKO mouse to evaluate Vangl2 regulation of actin treadmilling, N-cadherin dynamics and growth cone behavior.

Essential revisions:

1) The authors fail to reference or discuss several papers that appear relevant to Vangl2 function in axon guidance or outgrowth. The most significant omission from the Discussion is a manuscript by Love et al., 2018 which discusses Vangl2 mediated regulation of protrusions and cell migration which is analogous to growth cone outgrowth. However in this context Vangl2 function is fibronectin dependent and may not require N-Cadherin. How do these findings relate to each other and does this reflect a broader function for Vangl2 and cell adhesion molecules? Vangl2 also regulates growth cone migration through a non-autonomous mechanism (Ghimire, 2018). How could the cell-autonomous and non-autonomous functions for Vangl2 function to each other? Is this a common mechanism or multiple?

2) It is not clear whether the authors view this mechanism as a unique function for Vangl2, acting independent of other Core PCP factors, or whether N-Cadherin stabilization is an output of PCP signaling. This would be worth discussing as the authors imply that 'PCP signaling effects' this process (Discussion) but do not demonstrate a role for other PCP proteins such as Frizzled or CELSR that contribute to intercellular PCP signaling.

3) Related to point #2, could the authors discuss whether the presence of Vangl2 and possibly PCP polarization of the growth cone is contributing to an N-Cadherin mediated axon guidance (turning) events? Or is this just outgrowth?

4) The series of experiments carried out in vitro, including FRET experiments, single molecule tracking and their in-depth analyses/modeling set a new state-of-the art and understanding of the phenotypes. This is in sharp contrast with the very limited description of the in vivo conditional mutant phenotype presented in Figure 1. While it is clear that this is not the core of the article, it would be important to present a more precise description of the phenotype, and discuss how it might be related to an enhanced axonal outgrowth. To this aim, the authors could show horizontal sections through the corpus callosum and hippocampal commissure to assess the severity of the deficit along the rostrocaudal axis. In addition, they should explore or discuss whether cortical axons of the internal capsule, which progress during embryogenesis, show an accelerated outgrowth in Vangl2 conditional mutants using either classical immunostaining or axonal tracing.

5) Given the effect of the Loop-tail (Lp) mutation on cortical neurogenesis (precocious differentiation of neural progenitors into early-born neurons at the expense of late-born neurons; Lake and Sokol, 2009), and that the fact this might affect the size of the corpus callosum, the authors should test the effect of the loss of Vangl2 on cortical histogenesis, in particular on the production of upper-layer neurons.

---

## [Author Response]

Essential revisions:1) The authors fail to reference or discuss several papers that appear relevant to Vangl2 function in axon guidance or outgrowth. The most significant omission from the Discussion is a manuscript by Love et al., 2018 which discusses Vangl2 mediated regulation of protrusions and cell migration which is analogous to growth cone outgrowth. However in this context Vangl2 function is fibronectin dependent and may not require N-Cadherin. How do these findings relate to each other and does this reflect a broader function for Vangl2 and cell adhesion molecules?

We thank the reviewer for this remark that gives us the opportunity to clarify our view.

We are familiar with the work of J.R. Jessen and we agree that the work of that group highlight a link between core PCP signaling and specifically Vangl2, and ECM components in migrating cells, and as such should be mentioned. But we also believe there are many important mechanistic differences between the ectodermal cells of the zebrafish embryo during gastrulation and hippocampal neurons in the mouse, and we should be cautious in drawing direct mechanistic comparison between different systems and species.

The work of the Jessen’s group over the years demonstrated that Vangl2 can modulate the remodeling of the ECM during the gastrulation of zebrafish embryos, by regulating the endocytosis of the metalloproteinase MMP14 (matrix‐metalloproteinase). They showed that the loss of *Vangl2* in trilobite mutants leads to an increase in cell surface amounts of MMP14 and increased MMP14 proteolysis of fibronectin leading to a decrease in fibronectin levels, which is correlated to reduced cell–matrix adhesion and an increase in membrane protrusive activity. This was associated with disrupted polarized cell behavior (=PCP) and convergent extension defects (Williams et al., 2012). In 2013, the same group showed that cadherin levels were decreased at the plasma membrane in *Vangl2* mutant embryos (Dohn et al., 2013), and in 2017 that a loss of human VANGL2 reduced cell adhesion to fibronectin, laminin, and vitronectin in a cell line (Jessen and Jessen, 2017). The specific work mentioned by the reviewer (Love et al., 2018) suggests that fibronectin is necessary for Vangl2 recruitment at the plasma membrane of zebrafish ectodermal cells and that Vangl2 promotes protrusion in these cells. While the overall idea that core PCP proteins, and notably Vangl2, regulate plasma membrane proteins that will interfere with ECM molecules to modulate migratory behavior is common to both our systems (and others), they are very different in terms of mechanisms, including the regulation of cadherin levels, and the resulting adhesive and migratory properties.

In contrast with Jessen’s findings, we observe that a deletion of *Vangl2* in hippocampal neurons results in increased adhesion due to a reduced turnover of N-cadherin and in a resulting N-Cadherin-dependent (but not integrin/laminin dependent) regulation of axonal outgrowth via an increased physical link between the N-cadherin present in the substrate, the N-cadherin at the growth cone membrane and the cytoskeleton.

We believe these differences between systems emphasize the need to study the function of each core PCP gene individually in a specific cellular context, and that if these genes are evolutionary conserved, they may have developed new functions. Alternatively, they may have conserved similar function but with a different molecular basis through new binding partners, to adapt to a specific cellular context.

On the other hand, we believe that a common molecular theme to both systems (and others) could be a Vangl2-dependent regulation of endocytosis of proteins involved in adhesion remodeling, a point we develop more in response to question 2 below and that we now discuss in the Discussion.

The work of Jessen, including the Love et al., 2018, study as well as the Dohn et al., 2013, and the Jessen and Jessen, 2017, is now cited in the Introduction with other related citations, and in the Discussion section.

Vangl2 also regulates growth cone migration through a non-autonomous mechanism (Ghimire, 2018). How could the cell-autonomous and non-autonomous functions for Vangl2 function to each other? Is this a common mechanism or multiple?

While our in vitro data, in a very controlled environment, allowed the identification of a clear cell autonomous deficit due to the absence of Vangl2 in the neuron, in the absence of any other interfering bias, the in vivo phenotype observed in the Emx1**Vangl2* cKO is most probably the result of a combination of the disruption of both autonomous and non-autonomous Vangl2-dependent mechanisms. Vangl2 is broadly expressed in the cortex and hippocampus early in development, notably at the time of intense neurogenesis (Tissir et al., 2006) and its deletion could affect various mechanisms, in neuronal and non-neuronal cells. So, the environment explored by the growth cone as it moves might also be affected by *Vangl2* deletion, in terms, for example, of guideposts location/maturity, secretion of diffusible molecules, fasciculation or gene expression, including ECM expression levels. Our data on the redistribution of Vangl2 at the periphery of the growth cone on the N-cadherin substrate support the hypothesis that the composition of the ECM could reciprocally modulate Vangl2 molecules in the neuron. This highly dynamic interplay would allow the neuron to react to the ECM composition present along its path as it grows. We further discussed this in response to question 4, in the text below. This is further discussed in the Discussion section.

2) It is not clear whether the authors view this mechanism as a unique function for Vangl2, acting independent of other Core PCP factors, or whether N-Cadherin stabilization is an output of PCP signaling. This would be worth discussing as the authors imply that 'PCP signaling effects' this process (Discussion) but do not demonstrate a role for other PCP proteins such as Frizzled or CELSR that contribute to intercellular PCP signaling.

This is a fundamental question that goes back to some extent to our statement in the Discussion that “These results suggest that core PCP genes do not share a similar signaling cassette in young neurons as they do in epithelial cells, at least for axon guidance” (already present in the previous manuscript). We do believe that core PCP proteins can activate signaling pathways independent from each other. We believe this is the case for axonal guidance, where it was clearly shown that *Celsr3* and *Frizzled3* mutants have massive axonal bundle disruption in the forebrain while *Vangl2* and *Vangl1/Vangl2* double mutants do not.

Even in *Drosophila*, where all 6 core PCP genes are needed to control PCP in various epithelia, there are examples of neuronal signaling pathways that do not require all core PCP proteins. For example, one study suggests that during *Drosophila* midline guidance, Fmi and Fz act independently of the other classical PCP components, including Vang (Organisti et al., 2014). Also, a work in *C. elegans* on Vang‐1 and Prkl‐1 (*C. elegans* orthologous of Vang and Prickle proteins respectively) demonstrated that both proteins cooperated to negatively regulate neurite formation, without involvement of the other core PCP proteins (Sanchez‐Alvarez et al., 2011).

One the other hand, at the molecular level, it is tempting to conclude that Vangl2 has a global role as a regulator of endocytosis in the context of junctional remodeling. Vangl2 has been shown to interact with a large variety of proteins, many of which still await characterization (Bailly et al., 2018). Within these interactors, there is now accumulating evidence, in various species and tissues, that E-cadherin and N-cadherin turnover is modulated by Vangl2 and its orthologues, through direct or indirect interaction (Classen et al., 2005; Warrington et al., 2013; Nagaoka 2014a, b). Vangl2 has also been shown to regulate the endocytosis of other molecules including Frizzled 3 (Shafer et al., 2011), nephrin (Babayeva et al., 2013) or the matrix metalloproteinase MMP14 (Williams et al., 2012). Therefore, Vangl2-mediated membrane protein endocytosis might represent an evolutionarily conserved mechanism by which tissue morphogenesis (including neuronal tissue) is controlled in both vertebrates and invertebrates. This is now discussed in the revised Discussion section.

Also, the sentence referring globally to “PCP signaling effects” in the previous manuscript has been amended to “In summary, this work offers an original and dynamic view of the role of Vangl2 in neuronal cells and gives us insight into how a specific core PCP gene can affects the development and the morphogenesis of neuronal tissue.”

3) Related to point #2, could the authors discuss whether the presence of Vangl2 and possibly PCP polarization of the growth cone is contributing to an N-Cadherin mediated axon guidance (turning) events? Or is this just outgrowth?

It is not possible at this point to definitively conclude that Vangl2 is not at all involved in axonal guidance as defined here, as a turning event, at least in vivo. The response of one type of neuron, to one specific signal, is hard to predict in terms of integration of cues in a complex and changing environment. Our in vitro data, in a very controlled environment, allowed the identification of a clear cell autonomous deficit due to the absence of *Vangl2* in the neuron, in the absence of any other interfering bias. But even in vitro, there is an inherent complexity to the response of a neuron to many cues. For example, microfluidic analysis of hippocampal neurons in response to Slit1 showed that the diffusible molecule can be repulsive or attractive, depending if it is used alone, or in combination with a gradient of netrin-1, but also depending on the substrate (Dupin et al., 2015). A similar complexity was demonstrated regarding the response of retinal axons to netrin in the presence or absence of laminin (Hopker et al., 1999). One could imagine that in vivo, and in a subset of hippocampal neurons, guidance is regulated by the interplay between Vangl2 and the ratio of N-cadherin and laminin in the environment. When the growth cone encounters an N-cadherin-rich environment, Vangl2 could control the number of pauses (for example), allowing better exploration, and maybe influencing the orientation of the growth cone. But if this is the case, most of the forebrain neurons appear to be able to compensate for the absence of such a Vangl2 control of guidance since there is not a massive disruption of axonal guidance in the cKO.

Our in vivo results show that hippocampal neurons are still able to answer correctly to diffusible factors such as netrin-1 and Wnt5a in terms of axonal outgrowth when deprived of Vangl2. Based on these results, we suggest that given the correct cellular context, they might also turn correctly in response to these factors. Supporting this hypothesis, our in vivoresults show that axon guidance as a whole is not affected by *Vangl2* deletion, as we (and others) did not observe major abnormalities in the main forebrain axonal tracts, except for the absence of the hippocampal commissures (HC) and of the corpus callosum (CC) in the dorso-caudal area. This is in contrast to what was reported for Frizzled 3- and Celsr3-deficient neurons that seem to have normal outgrowth but do not respond to turning (Chai et al., 2014; Hua et al., 2014) and whose mutant brains display major forebrain axonal tract deficits. We believe that these differences in neuronal behavior and in the corresponding in vivo phenotypes, similar for Fzd3 and Celsr3 but different from Vangl2, suggest that the N-cadherin/Vangl2 relationship identified in our manuscript with hippocampal neurons mostly pertains to outgrowth. These points are now discussed in the Discussion section.

4) The series of experiments carried out in vitro, including FRET experiments, single molecule tracking and their in-depth analyses/modeling set a new state-of-the art and understanding of the phenotypes. This is in sharp contrast with the very limited description of the in vivo conditional mutant phenotype presented in Figure 1. While it is clear that this is not the core of the article, it would be important to present a more precise description of the phenotype, and discuss how it might be related to an enhanced axonal outgrowth. To this aim, the authors could show horizontal sections through the corpus callosum and hippocampal commissure to assess the severity of the deficit along the rostrocaudal axis. In addition, they should explore or discuss whether cortical axons of the internal capsule, which progress during embryogenesis, show an accelerated outgrowth in Vangl2 conditional mutants using either classical immunostaining or axonal tracing.

We thank the reviewer for his/her appreciation of our work and interest in further characterizing the Vangl2-dependent phenotype. It is true that the focus of this manuscript was to identify a molecular basis for the specific and restricted connectivity problems resulting from *Vangl2* deletion that are in contrast to what was reported for two other core PCP genes, *Frizzled3* and *Celsr3* (Wang et al., 2006; Qu et al., 2014, Chai et al., 2014, Hua et al., 2014).

To better illustrate the corpus callosum and hippocampal commissure deficits, we generated, as suggested by the reviewer, horizontal sections of 3-week-old mouse controls and mutants, showing in a single plane the presence of both the commissures in a ventral position and their absence in a more dorsal position. Some sections were processed for neurofilament staining, and other were stained with cresyl violet (new Figure 1B and Figure 1—figure supplement 1E). We also processed coronal sections from controls and mutants for cresyl violet stain and compiled them to generate a 3D rendering of corpus callosum and hippocampal structure + commissures in the dorsal region (new Figure 1D). This representation clearly illustrates that the CC deficit matches exactly that of the HC.

We also generated and illustrated a more rostral coronal section with fluoromyelin staining, illustrating the presence of the CC and HC in that location, in addition to the more caudal section that was presented in the previous manuscript where both commissures are missing (new Figure 1A). These new illustrations can be found in the new Figure 1 and Figure 1—figure supplement 1.

Regarding a potential difference in the growth of cortical axons from the internal capsule, we compared the growth of axons in embryos from E12.5, E13.5 and E14.5 control and cKO embryos after neurofilament staining. We did not observe a significant difference between controls and mutants. An illustration of E13.5 coronal sections from controls and cKOs labeled with neurofilament is now present in Figure 1—figure supplement 1C of the revised manuscript.

Regarding how the partial CC and HC agenesis might be related to an enhanced axonal outgrowth, it is obviously complicated to answer this without a more thorough in vivo analysis (including the deletion of *Vangl2* in specific neuronal population) that would be beyond the scope of this study. We can however make a few suggestions/comments.

Certainly, the in vivo phenotype observed in the Emx1**Vangl2* cKO is the result of a combination of the disruption of both autonomous and non-autonomous Vangl2-dependent mechanisms, as mentioned in response to question 3. Vangl2 is broadly expressed in the cortex and hippocampus early in development, notably at the time of intense neurogenesis (Tissir et al., 2006), and its deletion could affect various mechanisms in neuronal and non-neuronal cells. So, the environment explored by the growth cone as it moves might also be affected by *Vangl2* deletion, in terms for example of guideposts location/maturity, secretion of diffusible molecules and fasciculation or gene expression, including ECM expression levels. Our data on the redistribution of Vangl2 at the periphery of the growth cone on the N-cadherin substrate suggest that the composition of the ECM could reciprocally modulate Vangl2 molecules in the neuron. This highly dynamic interplay would allow the neuron to react to the ECM composition present along its path as it grows.

In this study, we have focused our analysis on hippocampal neurons in vitro, and we further suggest that the observed dorso-caudal CC deficit may be a consequence of the HC deficit. The CC crossing above the HC and the matching deficit are now clearly illustrated in the new Figure 1D. However, we cannot exclude that cortical neurons might also react abnormally to the absence of Vangl2, as *Vangl2* deletion might differentially affect different population of neurons as they encounter distinct environments. This has been illustrated for Fzd3 mutants both in terms of axonal growth and guidance, probably because of distinct responses to a specific diffusible cue (secreted or ECM) encountered as the axon grows (Hua et al., 2014).

All of these points are now discussed in the Discussion of the revised manuscript.

5) Given the effect of the Loop-tail (Lp) mutation on cortical neurogenesis (precocious differentiation of neural progenitors into early-born neurons at the expense of late-born neurons; Lake and Sokol, 2009), and that the fact this might affect the size of the corpus callosum, the authors should test the effect of the loss of Vangl2 on cortical histogenesis, in particular on the production of upper-layer neurons.

The additional data we generated following the reviewer’s comments show that despite an absence of Vangl2 protein in the dorsal telencephalon as early as E13.5 in Emx1**Vangl2* cKO (illustrated now in Figure 1—figure supplement 1B), no clear disruption of cortical thickness or of Cux1-positive cells (illustrated now Figure 1E-G) was observed, suggesting that the embryonic absence of Vangl2 in the dorsal telencephalon does not severely affect neuronal cell fate or migration. This is in sharp contrast with what was observed by Lake and Sokol, 2009, in a *Vangl2*^Lp/Lp^ brain. In that study, the authors observed a massive reduction of neocortex thickness that they linked to a disruption of oriented cell division in the progenitor pool of the ventricular zone. Such a discrepancy could be due to at least 3 reasons:

1) the Loop-tail mutation is a germline mutation and often these result in a stronger phenotype compared to conditional deletion. The severe neural tube defect (NTD) phenotype (craniorachischisis) (Kibar et al., 2010) observed in *Vangl2*^Lp/Lp^ or in a null *Vangl2*^*Δ/Δ*^ embryos (Song et al., 2010) is not observed in the Emx1**Vangl2* cKO (this manuscript) or in the double Vangl1^−/−^;*Foxg1Cre;Vangl2*^f/f^ cKO generated by Qu et al., 2014.

2) some of the central nervous system defects seen in *Vangl2*^Lp/Lp^ or in *Vangl2*^*Δ/Δ*^ embryos brains could also be a secondary consequence of these severe disruption of tissue morphogenesis as observed in the *Vangl2*^Lp/Lp^ or in a *Vangl2^*Δ/Δ*^* embryos but not in the neuronal-specific conditional mutant.

3) a final compounding reason of such a severe phenotype in *Vangl2*^Lp/Lp^ embryo could be the existence of a dominant-negative or a gain-of-function property of the Vangl2^Lp^ protein.

It is therefore probable that at least some of the severe phenotypes observed by Lake and Sokol is due to a deficit in very early mechanisms of neural tube closure (before E9.5) and involving mesenchymal and/or ectodermal cell movements required for closing of the neural tube. This does not disprove a role for Vangl2 in asymmetric cell division, but strongly suggests that the severity of the brain phenotype as observed in the *Vangl2*^Lp/Lp^ is probably not due solely to this process. This emphasizes again the caution necessary when drawing mechanistic conclusions on the role of Vangl2 (or Vangl1) when using the Loop-tail model.

These results have been added to Figure 1, and the corresponding discussion regarding the Lake and Sokol article can be found in the Discussion of the revised manuscript.

Additional references:

Dupin I, Lokmane L, Dahan M, Garel S, Studer V. Subrepellent doses of Slit1 promote Netrin-1 chemotactic responses in subsets of axons. Neural Dev. 2015 Mar 20;10:5.

Höpker VH, Shewan D, Tessier-Lavigne M, Poo M, Holt C. Growth-cone attraction to netrin-1 is converted to repulsion by laminin-1. Nature. 1999 Sep 2;401(6748):69-73

Kibar Z, Vogan KJ, Groulx N, Justice MJ, Underhill DA, Gros P. Ltap, a mammalian homolog of Drosophila Strabismus/Van Gogh, is altered in the mouse neural tube mutant Loop-tail. Nat Genet. 2001 Jul;28(3):251-5.

Organisti C, Hein I, Grunwald Kadow IC, Suzuki T. Flamingo, a seven-pass transmembrane cadherin, cooperates with Netrin/Frazzled in Drosophila midline guidance. Genes Cells. 2015 Jan;20(1):50-67

Sanchez‐Alvarez, L., Visanuvimol, J., McEwan, A., Su,A., Imai, J.H., and Colavita, A. (2011). VANG‐1 and PRKL‐Cooperate to Negatively Regulate Neurite Formation in Caenorhabditis elegans. PLOS Genet 7, e1002257